



# Technical note: Analytical sensitivity analysis and uncertainty estimation of a two-component hydrograph separation method which uses conductivity as a tracer

Weifei Yang[1], Changlai Xiao[1], Xiujuan Liang[1]

1 Key Laboratory of Groundwater Resources and Environment, Ministry of Education, and National-Local Joint Engineering Laboratory of In-situ Conversion, Drilling and Exploitation Technology for Oil Shale, and College of New Energy and Environment, Jilin University, No 2519, Jiefang Road, Changchun 130021, PR China

*Correspondence to:* Changlai Xiao (xcl2822@126.com, jluywf@126.com )

**Abstract.** The conductivity two-component hydrograph separation method is cheap and easy to operate and is favored by

hydrologists. This paper analyzes the sensitivity of the baseflow index (BFI, the long-term ratio of baseflow to streamflow) calculated by this method to errors or uncertainties of the two parameters ($BF_C$, the conductivity of baseflow; $RO_C$, the conductivity of surface runoff) and of the two variables ($y_k$, the specific streamflow; $Q_{ck}$, the specific conductivity of streamflow), and then estimates the uncertainty of BFI. The analysis shows that when the time series is longer than 365 days, the random measurement errors of $y_k$ or $Q_{ck}$ will cancel each other, and the influence on BFI can be neglected. Dimensionless

sensitivity indices (the ratio of the relative error of BFI to the relative error of $BF_C$ or $RO_C$) can well express the propagation of errors or uncertainties of $BF_C$ or $RO_C$ into BFI. Based on the sensitivity analysis, the uncertainty estimation method of BFI is derived. Representative sensitivity indices and BFI' uncertainties are yielded by application of the resulting equations to 24 watersheds in the United States. The results indicate that BFI is more sensitive to $BF_C$, and the conductivity two-component hydrograph separation method may be more suitable for the long time series in a small watershed. After considering the mutual

offset of the measurement errors of conductivity and streamflow, the uncertainty of BFI is reduced by half.

## 1 Introduction

Hydrograph separation (also called baseflow separation), can effectively identify the proportion of water in different runoff pathways in a basin's export flow, which helps to identify the conversion relationship between groundwater and surface water, and is a necessary condition for optimal allocation of water resources (Cartwright et al., 2014; Miller et al., 2014; Costelloe et al.,

2015). Chemical/isotope (tracer) hydrograph separation method is considered to be the most effective separation method, which can reflect the actual characteristics of a basin (Mei and Anagnostou, 2015; Zhang et al., 2017). Many hydrologists have indicated that electrical conductivity can be used as a tracer to perform hydrograph separation (Stewart et al., 2007; Munyaneza et al., 2012; Cartwright et al., 2014; Lott and Stewart, 2016; Okello et al., 2018). The measurement of conductivity is simple and inexpensive and has a distinct applicability in a long series of hydrograph separation (Okello et al., 2018).

The conductivity two-component hydrograph separation method (also called conductivity mass balance method (CMB) (Stewart et al. 2007)) uses conductivity as a tracer to calculate baseflow through a two-component mass balance equation. The general equation is shown in Eq. (1).

$$b_k = \frac{y_k(Q_{ck}-RO_C)}{BF_C-RO_C} \tag{1}$$

where $b$ is the baseflow ($L^3/t$), $y$ is the streamflow ($L^3/t$), $Q_c$ is the electrical conductivity of streamflow, and $k$ is the time step

number. The two parameters $BF_C$ and $RO_C$ respectively represent the electrical conductivity of baseflow and surface runoff.





The field test of Stewart et al. (2007) showed that the maximum conductivity of streamflow can be used to replace $BF_C$, and the minimum conductivity can be used to replace $RO_C$. However, Miller et al. (2014) pointed out that the maximum conductivity of streamflow may exceed the real $BF_C$, so they suggested that the 99th percentile of the conductivity of a long series of streamflow should be used as the $BF_C$ to avoid the impact of high $BF_C$ estimates on the separation results. There is uncertainty in

determining the parameters ($BF_C$, $RO_C$) of the conductivity two-component hydrograph separation method (Miller et al., 2014; Okello et al., 2018). The sensitivity analysis of parameters and the uncertainty quantitative analysis of separation results are helpful to further optimize the conductivity two-component hydrograph separation method and improve the accuracy of hydrograph separation.

Most of the existing parameter sensitivity analysis methods use experimental sensitivity analysis method, which usually

substitutes the fluctuation value of a certain parameter into the separation model, and then analyzes the sensitivity of the parameters by comparing the range of the separation results produced by these fluctuation parameters (Eckhradt, 2005; Miller et al., 2014; Okello et al., 2018). Eckhardt (2012) indicated that "An empirical sensitivity analysis is only an analytical calculation of the error propagation through the model, is not feasible." Eckhardt (2012) derived the sensitivity indices of the equation parameters by the partial derivative of a two-parameter recursive digital baseflow separation filter equation. However, the

parameters' sensitivity indices of the conductivity two-component hydrograph separation equation have not been derived.

At present, the uncertainty of the separation results of the conductivity two-component hydrograph separation method is mainly estimated by uncertainty transfer equation based on the uncertainty of $BF_C$, $RO_C$, and $Q_{ck}$ (Genereux, 1998; Miller et al., 2014). See Sect. 3.1 for details. This method can only estimate the uncertainty of the baseflow ratio ($f_{bf}$, the ratio of baseflow to streamflow in a single calculation process), and then use the average uncertainty of multiple calculation processes to estimate the

uncertainty of the baseflow index (BFI, the long-term ratio of baseflow to total streamflow). This uncertainty estimation method can neither directly estimate the uncertainty of BFI nor consider the randomness and mutual offset of conductivity measurement error, and the uncertainty estimation of BFI is not appropriate enough.

The purpose of this paper is to derive the parameters' sensitivity indices of the conductivity two-component hydrograph separation equation by calculating the partial derivative of Eq. (1) (Sect. 2), and further derive the direct estimation method of

BFI' uncertainty (Sect. 3). The derived methods were applied to 24 basins in the United States, and the parameters' sensitivity indices and BFI' uncertainty characteristics were analyzed (Sect. 4).

## 2 Analytical sensitivity analysis

### 2.1 Parameters $BF_C$ and $RO_C$

In order to calculate the sensitivity indices of the parameters, the partial derivatives of $b_k$ in Eq. (1) to $BF_C$ and $RO_C$ are required

respectively (for the derivation process, see Appendix A1, A2):

$$\frac{\partial b_k}{\partial BF_c} = -y_k \frac{Q_{ck}-RO_c}{(BF_c-RO_c)^2} \qquad (2)$$

$$\frac{\partial b_k}{\partial RO_c} = y_k \frac{Q_{ck}-BF_c}{(BF_c-RO_c)^2} \qquad (3)$$

For the convenience of comparison, the baseflow index (BFI) is selected as the baseflow separation result for long time series to analyze the influence of parameters' uncertainty on BFI,

$$BFI = \frac{\sum_{k=1}^{n} b_k}{\sum_{k=1}^{n} y_k} = \frac{b}{y} \qquad (4)$$

where $b$ denotes the total baseflow and $y$ the total streamflow over the whole available streamflow sequences, n is the number of available streamflow data.





Then, the partial derivatives of BFI to $BF_C$ and $RO_C$ should be calculated, (for the derivation process, see Appendix A3, A4):

$$\frac{\partial \mathrm{BFI}}{\partial \mathrm{BF_c}} = \frac{y\mathrm{RO_c} - \sum_{k=1}^{n} y_k Q_{ck}}{y(\mathrm{BF_c} - \mathrm{RO_c})^2} \tag{5}$$

$$\frac{\partial \mathrm{BFI}}{\partial \mathrm{RO_c}} = \frac{\sum_{k=1}^{n} y_k Q_{ck} - y\mathrm{BF_c}}{y(\mathrm{BF_c} - \mathrm{RO_c})^2} \tag{6}$$

It can be seen from the definition of the partial derivative that the influence of the errors of the parameters ($\Delta BF_C$ and $\Delta RO_C$) in Eq. (1) on the BFI can be expressed by the product of the errors and its partial derivatives. Then the BFI' errors caused by tiny errors of $BF_C$ and $RO_C$ can be expressed as:

$$\Delta_{\mathrm{BF_c}} \mathrm{BFI} = \frac{\partial \mathrm{BFI}}{\partial \mathrm{BF_c}} \Delta \mathrm{BF_c} = \frac{y\mathrm{RO_c} - \sum_{k=1}^{n} y_k Q_{ck}}{y(\mathrm{BF_c} - \mathrm{RO_c})^2} \Delta \mathrm{BF_c} \tag{7}$$

$$\Delta_{\mathrm{RO_c}} \mathrm{BFI} = \frac{\partial \mathrm{BFI}}{\partial \mathrm{RO_c}} \Delta \mathrm{RO_c} = \frac{\sum_{k=1}^{n} y_k Q_{ck} - y\mathrm{BF_c}}{y(\mathrm{BF_c} - \mathrm{RO_c})^2} \Delta \mathrm{RO_c} \tag{8}$$

The dimensionless sensitivity indices (S) can be obtained by comparing the relative error of BFI caused by the tiny errors of $BF_C$ and $RO_C$ with that of $BF_C$ and $RO_C$, (see Appendix B1, B2):

$$S(\mathrm{BFI}/\mathrm{BF_c}) = \frac{\Delta_{\mathrm{BF_c}} \mathrm{BFI}}{\mathrm{BFI}} \Big/ \frac{\Delta \mathrm{BF_c}}{\mathrm{BF_c}} = \frac{\mathrm{BF_c}(y\mathrm{RO_c} - \sum_{k=1}^{n} y_k Q_{ck})}{y\mathrm{BFI}(\mathrm{BF_c} - \mathrm{RO_c})^2} \tag{9}$$

$$S(\mathrm{BFI}/\mathrm{RO_c}) = \frac{\Delta_{\mathrm{RO_c}} \mathrm{BFI}}{\mathrm{BFI}} \Big/ \frac{\Delta \mathrm{RO_c}}{\mathrm{RO_c}} = \frac{\mathrm{RO_c}(\sum_{k=1}^{n} y_k Q_{ck} - y\mathrm{BF_c})}{y\mathrm{BFI}(\mathrm{BF_c} - \mathrm{RO_c})^2} \tag{10}$$

where $S(\mathrm{BFI}/\mathrm{BF_c})$ represent the dimensionless sensitivity index of BFI (output) with $BF_c$ (uncertain input), and $S(\mathrm{BFI}/\mathrm{RO_c})$ with $RO_c$.

The dimensionless sensitivity index is also called "elasticity index", it reflects the proportional relationship between the relative error of BFI and the relative error of parameters (e.g. $S(\mathrm{BFI}/\mathrm{BF_c}) = 1.5$, the relative error of $BF_c$ is 5%, then the relative error of BFI should be 1.5 times 5% (7.5%)). After determining the specific values of $BF_C$, $RO_C$, BFI, $y$, $y_k$ and $Q_{ck}$, the sensitivity indices $S(\mathrm{BFI}/\mathrm{BF_c})$ and $S(\mathrm{BFI}/\mathrm{RO_c})$ can be calculated and compared.

## 2.2 Variables $y_k$ and $Q_{ck}$

In addition to the two parameters, there are two variables ($Q_{ck}$ and $y_k$) in Eq. (1). This section will analyze the sensitivity of BFI to these two variables. Similar to Sect. 2.1, the partial derivatives of $b_k$ in Eq. (1) to $Q_{ck}$ and $y_k$ are obtained (see Appendix A5, A6), and the partial derivatives of BFI to $Q_{ck}$ and $y_k$ are further obtained (see Appendix A7, A8),

$$\frac{\partial \mathrm{BFI}}{\partial Q_{ck}} = \frac{1}{\mathrm{BF_c} - \mathrm{RO_c}} \tag{11}$$

$$\frac{\partial \mathrm{BFI}}{\partial y_k} = \frac{\sum_{k=1}^{n}(Q_{ck} - \mathrm{RO_c}) - \mathrm{BFI}(\mathrm{BF_c} - \mathrm{RO_c})}{y(\mathrm{BF_c} - \mathrm{RO_c})} \tag{12}$$

According to previous studies (Munyaneza et al., 2012; Cartwright et al., 2014; Miller et al., 2014; Okello et al., 2018) and this study (Table 1), the difference between $BF_C$ and $RO_C$ is often greater than 100 μs/cm, so $\partial \mathrm{BFI}/\partial Q_{ck}$ is usually less than 0.01. Appendix C shows that the value of $\partial \mathrm{BFI}/\partial y_k$ is usually far less than 1.

Tiny errors in $Q_{ck}$ and $y_k$ cause errors in BFI of

$$\Delta_{Q_{ck}} \mathrm{BFI} = \frac{\partial \mathrm{BFI}}{\partial Q_{ck}} \Delta Q_{ck} = \frac{\Delta Q_{ck}}{\mathrm{BF_c} - \mathrm{RO_c}} \tag{13}$$

$$\Delta_{y_k} \mathrm{BFI} = \frac{\partial \mathrm{BFI}}{\partial y_k} \Delta y_k = \frac{\sum_{k=1}^{n}(Q_{ck} - \mathrm{RO_c}) - \mathrm{BFI}(\mathrm{BF_c} - \mathrm{RO_c})}{y(\mathrm{BF_c} - \mathrm{RO_c})} \Delta y_k \tag{14}$$

The errors of BFI caused by $Q_{ck}$ and $y_k$ are summed up to get the error of BFI caused by $\sum_{k=1}^{n} Q_{ck}$ and $\sum_{k=1}^{n} y_k$ in the whole time series:

$$\Delta_{\sum_{k=1}^{n} Q_{ck}} \mathrm{BFI} = \sum_{k=1}^{n} \Delta_{Q_{ck}} \mathrm{BFI} = \sum_{k=1}^{n} \frac{\Delta Q_{ck}}{\mathrm{BF_c} - \mathrm{RO_c}} = \frac{1}{\mathrm{BF_c} - \mathrm{RO_c}} \sum_{k=1}^{n} \Delta Q_{ck} \tag{15}$$



$$\Delta_{\sum_{k=1}^{n} y_k} \text{BFI} = \sum_{k=1}^{n} \Delta_{y_k} \text{BFI} = \sum_{k=1}^{n} \left( \frac{\sum_{k=1}^{n}(Q_{ck}-\text{RO}_c) - \text{BFI}(\text{BF}_c-\text{RO}_c)}{y(\text{BF}_c-\text{RO}_c)} \Delta y_k \right) = \frac{\sum_{k=1}^{n}(Q_{ck}-\text{RO}_c) - \text{BFI}(\text{BF}_c-\text{RO}_c)}{y(\text{BF}_c-\text{RO}_c)} \sum_{k=1}^{n} \Delta y_k \quad (16)$$

The tiny errors in $Q_{ck}$ and $y_k$ are mainly composed of random analysis errors. Random errors mostly follow a normal distribution or a uniform distribution. The magnitude and direction of the random errors are usually not fixed. As the number of measurements increases, the positive and negative errors can compensate each other, and the average value of the errors will

gradually trend to zero (Huang and Chen, 2011).

The uncertainty of the instruments is <5% for $Q_{ck}$ less than 100 µs/cm and <3% for $Q_{ck}$ greater than 100 µs/cm (Wagner et al., 2006; Miller et al., 2014). The measurement uncertainty of streamflow is usually <3% (Zhang, 2005). In this paper, the error ranges of $Q_{ck}$ and $y_k$ are considered to be ±5% and ±3%, respectively. Considering the mutual offset of random errors, when the time series (n) is long enough, $\sum_{k=1}^{n} \Delta Q_{ck}$ in Eq. (15) and $\sum_{k=1}^{n} \Delta y_k$ in Eq. (16) will approach zero. Therefore, when n is large

enough, the error of BFI caused by the errors of $Q_{ck}$ and $y_k$ can be neglected.

To verify this phenomenon, the study collected the daily average conductivity and daily average streamflow of the surface water station with the USGS site number 0297100 (Table 1) from 2001 to 2010 (2979 days in total). Then, office Excel was used to generate 10 sets (2979 per set) of random numbers between -0.05 and 0.05 that obey normal distribution and uniform distribution respectively to simulate the errors (%) of the daily average conductivity. And 10 sets (2979 per set) of random numbers obeying

normal distribution and uniform distribution between -0.03 and 0.03, respectively, were used to simulate the errors (%) of the daily average streamflow. Finally, according to different time series (n) (e.g. 30, 60, 90, 120, 150, 180, 210, 240, 270, 300, 365, 730, 1095, … , 2979, days) sum the errors value ($\sum_{k=1}^{n} \Delta Q_{ck}$ and $\sum_{k=1}^{n} \Delta y_k$) and analyze the trend of the average error (%) with n.

The trend of the average error (%) of conductivity with n is shown in Fig. 1. The average errors of the uniform distribution (Fig.

1(a)) and the normal distribution (Fig. 1(b)) are all gradually approach zero with the increase of the time series (n), and the uniform distribution converges faster than the normal distribution. The average errors of the two distributions are between -2% and 2%, and the absolute value of the average errors are less than 0.49% when n is greater than 365.

Similar to the conductivity, the trend of the average error (%) of the streamflow with n is shown in Fig. 2. The average errors of the uniform distribution (Fig. 2(a)) and the normal distribution (Fig. 2(b)) all gradually approach to zero as the time series (n)

increases, and the uniform distribution converges faster than the normal distribution. The average errors of different n under the two distributions are between -2% and 2%, and the absolute value of the average errors are less than 0.67% when n is greater than 365.

From the above analysis, when the time series (n) is greater than 365 days (1 year), $\Delta_{\sum_{k=1}^{n} Q_{ck}} \text{BFI}$ will be less than 0.0049% (0.01 times 0.49%), and $\Delta_{\sum_{k=1}^{n} y_k} \text{BFI}$ will be much less than 0.76% (1 times 0.76%). Therefore, the random errors of daily average

conductivity and streamflow have a negligible effect on BFI.

**Figure 1. Average conductivity error (%) with different distributions along the time series (n), (a) uniform distribution, (b) normal distribution.**

**Figure 2. Average streamflow error (%) with different distributions along the time series (n), (a) uniform distribution, (b) normal distribution.**





## 3 Uncertainty estimation

### 3.1 Previous attempts

According to previous studies, in the case where a parameter $g$ is calculated as a function of several factors $x_1$, $x_2$, $x_3$, ..., $x_n$ (e.g. $g$= G($x_1$, $x_2$, $x_3$, ..., $x_n$)). The transfer equation between the uncertainty of the independent factors and the uncertainty of $g$ is

(Taylor, 1982; Kline, 1985; Genereux, 1998):

$$W_g = \sqrt{(\frac{\partial g}{\partial x_1} W_{x_1})^2 + (\frac{\partial g}{\partial x_2} W_{x_2})^2 + \cdots + (\frac{\partial g}{\partial x_n} W_{x_n})^2} \tag{17}$$

where $W_g$, $W_{x1}$, $W_{x2}$, and $W_{xn}$ are the same type of uncertainty values (e.g. all average errors or all standard deviations) for $g$, $x_1$, $x_2$, and $x_n$, respectively.

Based on the above principle, Genereux (1998) substituted Eq. (18) into Eq. (17) to derive the uncertainty estimation equation

(Eq. (19)) of the two-component mass balance baseflow separation method:

$$f_{bf} = \frac{Q_{ck} - \text{RO}_c}{\text{BF}_c - \text{RO}_c} \tag{18}$$

$$W_{f_{bf}} = \sqrt{(\frac{f_{bf}}{\text{BF}_c - \text{RO}_c} W_{\text{BF}_C})^2 + (\frac{1 - f_{bf}}{\text{BF}_c - \text{RO}_c} W_{\text{RO}_c})^2 + (\frac{1}{\text{BF}_c - \text{RO}_c} W_{Q_c})^2} \tag{19}$$

where $f_{bf}$ is the ratio of baseflow to streamflow in a single calculation process, $W_{fbf}$ is the uncertainty in $f_{bf}$ at the 95% confidence interval, $W_{\text{BFC}}$ is the standard deviation of the highest 1% of measured conductivity multiplied by the t-value ($\alpha$=0.05; two-tail) from the Student's distribution, $W_{\text{ROC}}$ is the standard deviation of the lowest 1% of measured conductivity multiplied by the t-

value ($\alpha$=0.05; two-tail) from the Student's distribution, and $W_{\text{QC}}$ is the analytical error in the conductivity multiplied by the t-value ($\alpha$=0.05; two-tail) (Miller et al., 2014).

Equation (19) can better estimate the uncertainty of $f_{bf}$ within a single calculation step. Hydrologists usually estimate the uncertainty of BFI approximately by averaging the uncertainty of all steps (Genereux, 1998; Miller et al., 2014). However, this

method does not consider the mutual offset of the conductivity measurement errors, and cannot accurately reflect the uncertainty of BFI. In this paper, based on the parameter sensitivity analysis, the uncertainty estimation equation of BFI is derived. See the next section for details.

### 3.2 Uncertainty estimation in BFI

BFI is a function of $\text{BF}_c$, $\text{RO}_c$, $Q_{ck}$ and $y_k$. And the uncertainty of $\text{BF}_c$, $\text{RO}_c$, $Q_{ck}$ and $y_k$ is independent of each other. Sect. 2.2

has explained that the random errors of daily average conductivity and streamflow have a negligible effect on BFI when the time series (n) is greater than 365 days (1 year), so the uncertainty of BFI can be expressed as:

$$W_{BFI} = \sqrt{(\frac{\partial \text{BFI}}{\partial \text{BF}_c} W_{\text{BF}_C})^2 + (\frac{\partial \text{BFI}}{\partial \text{RO}_c} W_{\text{RO}_C})^2} \tag{20}$$

where (see Eq. 5 and Eq. 9; Eq. 6 and Eq. 10)

$$\frac{\partial \text{BFI}}{\partial \text{BF}_c} = S(\text{BFI}/\text{BF}_c) \frac{\text{BFI}}{\text{BF}_c} \tag{21}$$

$$\frac{\partial \text{BFI}}{\partial \text{RO}_c} = S(\text{BFI}/\text{RO}_c) \frac{\text{BFI}}{\text{RO}_c} \tag{22}$$

Then, the Eq. (20) can be rewritten as:

$$W_{BFI} = \sqrt{(S(\text{BFI}/\text{BF}_c) \frac{\text{BFI}}{\text{BF}_c} W_{\text{BF}_C})^2 + (S(\text{BFI}/\text{RO}_c) \frac{\text{BFI}}{\text{RO}_c} W_{\text{RO}_C})^2} \tag{23}$$

where $W_{\text{BFI}}$, $W_{\text{BFC,}}$ and $W_{\text{ROC}}$ are the same type of uncertainty values for BFI, $\text{BF}_C$, and $\text{RO}_C$, respectively. For instance, $W_{\text{BFI}}$ is the uncertainty in BFI at the 95% confidence interval, $W_{\text{BFC}}$ is the standard deviation of the highest 1% of measured conductivity





multiplied by the t-value (α=0.05; two-tail) from the Student's distribution, $W_{ROC}$ is the standard deviation of the lowest 1% of measured conductivity multiplied by the t-value (α=0.05; two-tail) from the Student's distribution.

## 4 Application

### 4.1 Data and processing

The above sensitivity analysis and uncertainty estimation methods were applied to 24 catchments in the United States (Table 1). All basins used in this study are perennial streams, with drainage areas ranging from 10 km² to 1258481 km². Each gage has about at least 1 year of continuous streamflow and conductivity at the same period. All streamflow and conductivity data are daily average values retrieved from the United States Geological Survey's (USGS) National Water Information System (NWIS) website, http://waterdata.usgs.gov/nwis.

The daily baseflow of each basin was calculated using Eq. (1). The 99th percentile of the conductivity of the whole series of streamflow in each basin was used as the $BF_C$ and the 1st percentile as the $RO_C$. The total baseflow $b$, the total streamflow $y$ and the baseflow index BFI of each watershed were then calculated. According to the results of the hydrograph separation, the parameter sensitivity indices of BFI for $BF_C$ ($S(BFI/BF_c)$) and $RO_C$ ($S(BFI/RO_c)$) were calculated by Eq. (9) and Eq. (10), respectively.

Finally, the uncertainty of $f_{bf}$ in each step was calculated by Eq. (19) and averaged to obtain the Mean $W_{fbf}$ in each basin. The uncertainty ($W_{BFI}$) of BFI was directly calculated by Eq. (23), and then the values of Mean $W_{fbf}$ and $W_{BFI}$ were compared. For each basin, $W_{BFC}$ is the standard deviation of the highest 1% of measured conductivity multiplied by the t-value (α=0.05; two-tail) from the Student's distribution, $W_{ROC}$ is the standard deviation of the lowest 1% of measured conductivity multiplied by the t-value (α=0.05; two-tail) from the Student's distribution, and $W_{QC}$ is the analytical error in the conductivity (5%) multiplied by

the t-value (α=0.05; two-tail).

### 4.2 Results and discussion

The calculation results are shown in Table 1. The average baseflow index of the 24 watersheds is 0.29, the average sensitivity index of BFI for $BF_C$ ($S(BFI/BF_c)$) is -1.39, and the average sensitivity index of BFI for $RO_C$ ($S(BFI/RO_c)$) is -0.98. The negative sensitivity indices indicate a negative correlation between BFI and $BF_C$, $RO_C$. The sensitivity index for $BF_C$ is generally

greater than that for $RO_C$, indicating that BFI is more affected by $BF_C$ (for example, there are 5% uncertainty in both $BF_C$ and $RO_C$, then $BF_C$ leads to -1.39 times of uncertainty in BFI (-6.95%), while $RO_C$ leads to -0.98 times (4.9%)). Therefore, the determination of $BF_C$ requires more caution, and any small error may lead to greater uncertainty in BFI. Miller et al. (2014) have indicated that anthropogenic activities over long periods of time, or year to year changes in the elevation of the water table may result in temporally changing in the $BF_C$. He recommended taking different $BF_C$ values per year based on the conductivity values

at low flow periods to avoid the effects of $BF_C$' temporally fluctuations.

**Table 1. Basic information, parameter sensitivity analysis, and uncertainty estimation results for 24 basins in the United States. Footnote "a" in the "Area" column indicates that the values are estimated based on data from adjacent sites.**

The sensitivity index of BFI for $BF_C$ has a decreasing trend with the increase of time series (n) (Fig. 3(a)) and has an increasing

trend with the increase of watershed area (Fig. 3(b)), the correlation coefficients are 0.1698 and 0.4468, respectively. Although the correlations are not obvious, it still has important guiding significance. In the large basin, there are many different subsurface flow paths contributing to stream (Okello et al., 2018), each of which has a unique conductivity value (Miller et al., 2014). It is





difficult to represent the conductivity characteristics of subsurface flow with a special value. Therefore, the conductivity two-component hydrograph separation method has a higher applicability in a small watershed of long time series.

The sensitivity index of BFI for $RO_C$ did not change significantly with the increase of time series and watershed area (Fig. 3(c), Fig. 3(d)). During the rainstorm, the water level of the stream rises sharply, the subsurface flow is suppressed, and the
streamflow is almost entirely from the rainfall runoff. At this time, the conductivity of the stream is similar to the conductivity of the local rainfall (Stewart et al., 2007). The electrical conductivity of regional rainfall varies slightly, usually at a fixed value, and has no significant relationship with basin area and year (Munyaneza et al., 2012). Therefore, the temporal and spatial variation characteristics of BFI for $RO_C$ are not obvious.

**Figure 3. Scatter plots of sensitivity indices vs. time series (n) and drainage area of the 24 US basins. The watershed area uses a logarithmic axis, while the others are normal axes**.

Genereux's method (Eq.19) estimates the average uncertainty of BFI in the 24 basins (Average of Mean $W_{fbf}$) to be 0.13, whereas the average uncertainty of BFI (Average of $W_{BFI}$) calculated directly by this paper' method (Eq. 23) is 0.06 (Table 1). Mean $W_{fbf}$ in each basin is generally larger than $W_{BFI}$ ($W_{BFI}$ is about 0.51 times of Mean $W_{fbf}$), and there is a significant linear correlation
(Fig. 4). This shows that the two methods have the same volatility characteristics for BFI uncertainty estimation results, but Genereux's method (Eq. 19) often overestimates the uncertainty of BFI. This also means that when the time series is longer than 365 days (1 year), the measurement errors of conductivity and streamflow will cancel each other and thus reduce the uncertainty of BFI (about half of the original).

**Figure 4. Scatter plot of uncertainty in BFI ($W_{BFI}$) and mean uncertainty in $f_{bf}$ (Mean $W_{fbf}$).**

**5 Conclusions**

Equation (9) and Eq. (10) can well calculate the sensitivity indices of BFI for $BF_C$ and $RO_C$. Eq. (23) can estimate the uncertainty of BFI when the time series is larger than 365 days, taking into account the mutual cancellation of conductivity measurement errors. Applications in 24 basins in the United States showed that BFI is more sensitive to $BF_C$, and future studies should devote
more effort to determining the value of $BF_C$. In addition, the conductivity two-component hydrograph separation method may be more suitable for the long time series in a small watershed.

When the time series is greater than 365 days, the measurement errors of conductivity and streamflow have obvious mutual offset, and its influence on BFI can be neglected. After considering the mutual offset of random errors, the uncertainty of BFI will be reduced to half.

The above conclusions are only from the average of the 24 basins in the United States, and further research is needed in other countries or in more watersheds. The research in this paper only focuses on the two-component hydrograph separation method with conductivity as a tracer, but the parameter sensitivity analysis and uncertainty analysis method of other tracers are very similar to this paper, and it is easy to derive similar equations.

**Appendix A**

Calculation of the partial derivatives

$$\frac{\partial b_k}{\partial BF_c} = \frac{\partial}{\partial BF_c} \frac{y_k(Q_{ck} - RO_c)}{BF_c - RO_c} = y_k(Q_{ck} - RO_c)\frac{\partial}{\partial BF_c}\frac{1}{BF_c - RO_c} = -y_k\frac{Q_{ck} - RO_c}{(BF_c - RO_c)^2} \tag{A1}$$





$$\frac{\partial b_k}{\partial RO_c} = \frac{\partial}{\partial RO_c}\frac{y_k(Q_{ck}-RO_c)}{BF_c-RO_c} = y_k\frac{\partial}{\partial RO_c}\frac{Q_{ck}-RO_c}{BF_c-RO_c} = y_k\frac{-(BF_c-RO_c)+(Q_{ck}-RO_c)}{(BF_c-RO_c)^2} = y_k\frac{Q_{ck}-BF_c}{(BF_c-RO_c)^2} \tag{A2}$$

$$\frac{\partial BFI}{\partial BF_c} = \frac{\partial}{\partial BF_c}\frac{b}{y} = \frac{1}{y}\sum_{k=1}^{n}\frac{\partial b_k}{\partial BF_c} = \frac{1}{y}\sum_{k=1}^{n}(-y_k\frac{Q_{ck}-RO_c}{(BF_c-RO_c)^2})(\text{see Eq. A1}) = \frac{1}{y(BF_c-RO_c)^2}\sum_{k=1}^{n}(y_kRO_c-y_kQ_{ck}) = \frac{yRO_c-\sum_{k=1}^{n}y_kQ_{ck}}{y(BF_c-RO_c)^2} \tag{A3}$$

$$\frac{\partial BFI}{\partial RO_c} = \frac{\partial}{\partial RO_c}\frac{b}{y} = \frac{1}{y}\sum_{k=1}^{n}\frac{\partial b_k}{\partial RO_c} = \frac{1}{y}\sum_{k=1}^{n}(y_k\frac{Q_{ck}-BF_c}{(BF_c-RO_c)^2})(\text{see Eq. A2}) = \frac{1}{y(BF_c-RO_c)^2}\sum_{k=1}^{n}(y_kQ_{ck}-y_kBF_c) = \frac{\sum_{k=1}^{n}y_kQ_{ck}-yBF_c}{y(BF_c-RO_c)^2} \tag{A4}$$

$$\frac{\partial b_k}{\partial Q_{ck}} = \frac{\partial}{\partial Q_{ck}}\frac{y_k(Q_{ck}-RO_c)}{BF_c-RO_c} = \frac{1}{BF_c-RO_c}\frac{\partial}{\partial Q_{ck}}y_k(Q_{ck}-RO_c) = \frac{y_k}{BF_c-RO_c} \tag{A5}$$

$$\frac{\partial b_k}{\partial y_k} = \frac{\partial}{\partial y_k}\frac{y_k(Q_{ck}-RO_c)}{BF_c-RO_c} = \frac{(Q_{ck}-RO_c)}{BF_c-RO_c}\frac{\partial}{\partial y_k}y_k = \frac{Q_{ck}-RO_c}{BF_c-RO_c} \tag{A6}$$

$$\frac{\partial BFI}{\partial Q_{ck}} = \frac{\partial}{\partial Q_{ck}}\frac{b}{y} = \frac{1}{y}\sum_{k=1}^{n}\frac{\partial b_k}{\partial Q_{ck}} = \frac{1}{y}\sum_{k=1}^{n}\frac{y_k}{BF_c-RO_c}(\text{see Eq. A5}) = \frac{1}{y(BF_c-RO_c)}\sum_{k=1}^{n}y_k = \frac{1}{BF_c-RO_c} \tag{A7}$$

$$\frac{\partial BFI}{\partial y_k} = \frac{\partial}{\partial y_k}\frac{b}{y} = \frac{\partial}{\partial y_k}\frac{\sum_{k=1}^{n}b_k}{\sum_{k=1}^{n}y_k} = \frac{(\sum_{k=1}^{n}b_k)'(\sum_{k=1}^{n}y_k)-(\sum_{k=1}^{n}b_k)(\sum_{k=1}^{n}y_k)'}{(\sum_{k=1}^{n}y_k)^2} = \frac{y(\sum_{k=1}^{n}b_k)'-b(\sum_{k=1}^{n}y_k)'}{y^2} = \frac{y\sum_{k=1}^{n}(\frac{Q_{ck}-RO_c}{BF_c-RO_c})-b}{y^2}(\text{see Eq. A6}) =$$

$$\frac{y\sum_{k=1}^{n}(Q_{ck}-RO_c)-b(BF_c-RO_c)}{y^2(BF_c-RO_c)} = \frac{\sum_{k=1}^{n}(Q_{ck}-RO_c)-BFI(BF_c-RO_c)}{y(BF_c-RO_c)} \tag{A8}$$

## Appendix B

Calculation of the sensitivity indices

$$S(BFI/BF_c) = \frac{\Delta_{BF_c}BFI}{BFI}/\frac{\Delta BF_c}{BF_c} = \frac{yRO_c-\sum_{k=1}^{n}y_kQ_{ck}}{y(BF_c-RO_c)^2}\Delta BF_c\frac{BF_c}{BFI\Delta BF_c}(\text{see Eq. 7}) = \frac{BF_c(yRO_c-\sum_{k=1}^{n}y_kQ_{ck})}{yBFI(BF_c-RO_c)^2} \tag{B1}$$

$$S(BFI/RO_c) = \frac{\Delta_{RO_c}BFI}{BFI}/\frac{\Delta RO_c}{RO_c} = \frac{\sum_{k=1}^{n}y_kQ_{ck}-yBF_c}{y(BF_c-RO_c)^2}\Delta RO_c\frac{RO_c}{BFI\Delta RO_c}(\text{see Eq. 8}) = \frac{RO_c(\sum_{k=1}^{n}y_kQ_{ck}-yBF_c)}{yBFI(BF_c-RO_c)^2} \tag{B2}$$

## Appendix C

Prove that $\partial BFI/\partial y_k$ is far less than 1

$$\frac{\partial BFI}{\partial y_k} = \frac{\sum_{k=1}^{n}(Q_{ck}-RO_c)-BFI(BF_c-RO_c)}{y(BF_c-RO_c)}(\text{see Eq. A8}) \tag{C1}$$

Because of BFI>0, $(BF_c-RO_c)>0$, the above formula can be simplified:

$$\frac{\partial BFI}{\partial y_k} < \frac{\sum_{k=1}^{n}(Q_{ck}-RO_c)}{y(BF_c-RO_c)} \tag{C2}$$

Since $BF_c$ is usually much larger than $Q_{ck}$, the above formula can be rewritten as:

$$\frac{\partial BFI}{\partial y_k} < \frac{\sum_{k=1}^{n}(BF_c-RO_c)}{y(BF_c-RO_c)} = \frac{n(BF_c-RO_c)}{y(BF_c-RO_c)} = \frac{n}{y} = \frac{1}{\bar{y}} \tag{C3}$$

The daily average streamflow ($\bar{y}$) is usually much larger than 1 m$^3$/d, so $\partial BFI/\partial y_k$ is far less than 1.

*Data availability*

All streamflow and conductivity data can be retrieved from the United States Geological Survey's (USGS) National Water Information System (NWIS) website use the special gage number, http://waterdata.usgs.gov/nwis.




*Author contributions*

Weifei Yang, Changlai Xiao and Xiujuan Liang designed the research train of thought. Weifei Yang and Changlai Xiao completed the parameters' sensitivity analysis. Xiujuan Liang completed the uncertainty estimate of BFI. Weifei Yang carried out most of the data analysis and prepared the manuscript with contributions from all co-authors.

*Competing interests*

The authors declare that they have no conflict of interest.

*Acknowledgements*

This work is supported by the National Natural Science Foundation of China (41572216), the Provincial School Co-construction Project Special -- Leading Technology Guide (SXGJQY2017-6), the China Geological Survey Shenyang Geological Survey
Center "Changji Economic Circle Geological Environment Survey" project (121201007000150012), and the Jilin Province Key Geological Foundation Project (2014-13). We thank the anonymous reviewers for useful comments to improve the manuscript.

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

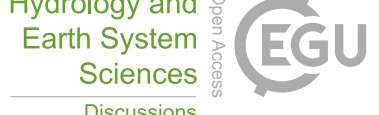



**Tables**

**Table 1. Basic information, parameter sensitivity analysis, and uncertainty estimation results for 24 basins in the United States. Footnote "a" in the "Area" column indicates that the values are estimated based on data from adjacent sites.**

| State | Gage Number | N | Area | $BF_C$ | $RO_C$ | Mean Baseflow | BFI | $S(BFI/BF_C)$ | $S(BFI/RO_C)$ | $W_{BFI}$ | Mean $W_{fbf}$ |
|-------|-------------|------|--------|--------|--------|--------|------|-------|-------|------|------|
|       |             | days | km² | µs/cm | µs/cm | m³/s |      |       |       |      |      |
| FL | 2298202 | 1808 | 966 | 1190.0 | 292.5 | 3.05 | 0.29 | -1.31 | -0.78 | 0.04 | 0.11 |
| FL | 2310545 | 1218 | 119[a] | 7150.5 | 531.5 | 0.10 | 0.15 | -1.09 | -0.44 | 0.05 | 0.06 |
| FL | 2310650 | 779 | 77[a] | 7195.0 | 3210.0 | 0.08 | 0.45 | -1.79 | -0.98 | 0.06 | 0.14 |
| FL | 2303000 | 728 | 570 | 462.0 | 120.5 | 3.28 | 0.30 | -1.30 | -0.82 | 0.08 | 0.17 |
| FL | 2298488 | 1303 | 76 | 810.0 | 194.0 | 0.20 | 0.33 | -1.30 | -0.63 | 0.05 | 0.09 |
| FL | 2298554 | 899 | 207[a] | 1155.0 | 320.5 | 0.25 | 0.20 | -1.36 | -1.55 | 0.03 | 0.08 |
| FL | 2298492 | 1478 | 16 | 1425.0 | 304.0 | 0.04 | 0.21 | -1.26 | -1.01 | 0.03 | 0.07 |
| FL | 2298495 | 330 | 10 | 1905.0 | 662.0 | 0.05 | 0.24 | -1.51 | -1.66 | 0.03 | 0.08 |
| FL | 2298527 | 807 | 23 | 1640.0 | 201.5 | 0.04 | 0.14 | -1.10 | -0.83 | 0.06 | 0.16 |
| FL | 2298530 | 1510 | 17 | 1520.0 | 348.0 | 0.13 | 0.27 | -1.27 | -0.80 | 0.07 | 0.12 |
| FL | 2297100 | 2979 | 342 | 1460.0 | 221.5 | 1.54 | 0.21 | -1.17 | -0.69 | 0.04 | 0.09 |
| FL | 2313000 | 787 | 4727 | 449.0 | 173.0 | 8.62 | 0.43 | -1.62 | -0.84 | 0.06 | 0.13 |
| FL | 2300500 | 821 | 386 | 470.0 | 83.0 | 0.49 | 0.19 | -1.19 | -0.90 | 0.11 | 0.20 |
| ND | 5057000 | 1401 | 16757 | 1520.0 | 610.0 | 1.73 | 0.46 | -1.64 | -0.81 | 0.09 | 0.15 |
| ND | 5056000 | 1277 | 5361 | 1770.0 | 546.0 | 2.50 | 0.42 | -1.41 | -0.61 | 0.04 | 0.11 |
| TX | 8068275 | 2801 | 482 | 368.0 | 65.0 | 4.20 | 0.15 | -1.18 | -1.23 | 0.06 | 0.13 |
| GA | 2336300 | 1235 | 225 | 230.0 | 63.0 | 4.00 | 0.29 | -1.36 | -0.93 | 0.24 | 0.42 |
| GA | 2207120 | 1383 | 417 | 381.0 | 59.0 | 3.97 | 0.18 | -1.17 | -0.86 | 0.03 | 0.06 |
| SC | 2160105 | 1363 | 1966 | 150.0 | 51.0 | 40.27 | 0.25 | -1.49 | -1.56 | 0.03 | 0.10 |
| SC | 2160700 | 1392 | 1150 | 181.0 | 51.0 | 24.02 | 0.26 | -1.37 | -1.13 | 0.05 | 0.11 |
| MO | 6894000 | 1375 | 477 | 1110.0 | 334.0 | 0.86 | 0.21 | -1.40 | -1.59 | 0.09 | 0.13 |
| MO | 6895500 | 802 | 1258481 | 800.0 | 428.0 | 904.39 | 0.55 | -2.14 | -0.95 | 0.05 | 0.18 |
| ND | 5082500 | 1274 | 77959 | 1670.0 | 427.0 | 41.48 | 0.27 | -1.33 | -0.95 | 0.06 | 0.09 |
| KS | 7144780 | 575 | 1847 | 1550.0 | 678.0 | 0.52 | 0.44 | -1.60 | -1.08 | 0.09 | 0.17 |
| | | | Mean | | | | 0.29 | -1.39 | -0.98 | 0.06 | 0.13 |
| | | | Standard deviation (STDEV) | | | | 0.11 | 0.24 | 0.32 | 0.04 | 0.07 |





**Figures**

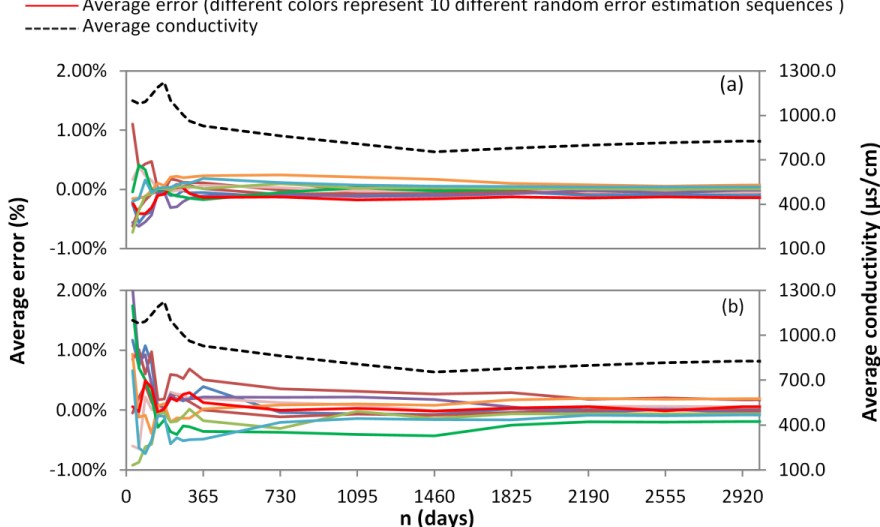

**Figure 1. Average conductivity error (%) with different distributions along the time series (n), (a) uniform distribution, (b) normal distribution.**





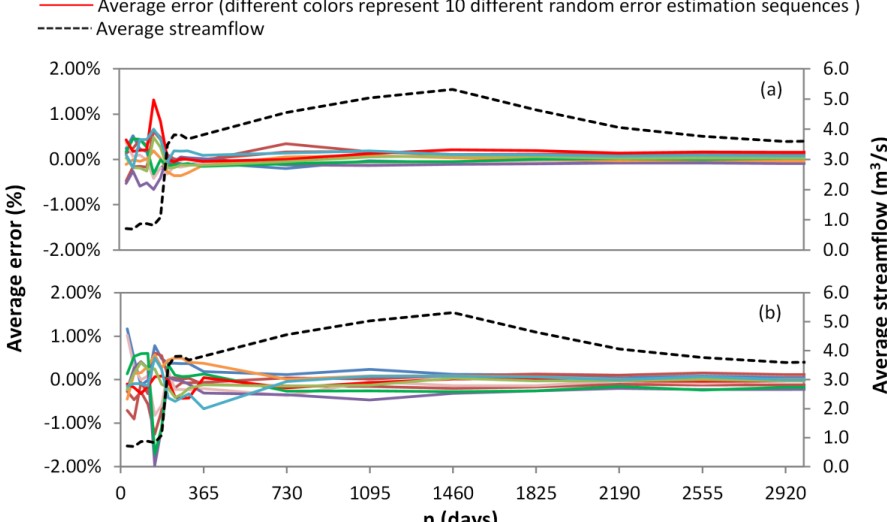

**Figure 2. Average streamflow error (%) with different distributions along the time series (n), (a) uniform distribution, (b) normal distribution.**





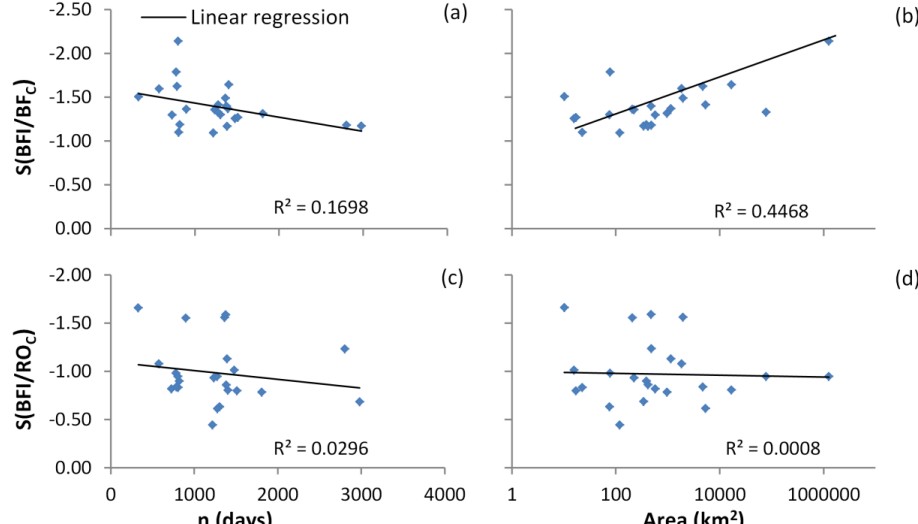

**Figure 3. Scatter plots of sensitivity indices vs. time series (n) and drainage area of the 24 US basins. The watershed area uses a logarithmic axis, while the others are normal axes**.



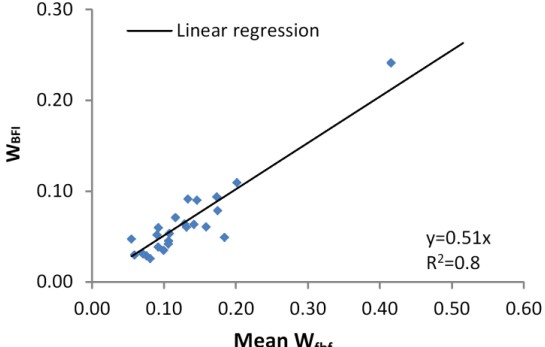

**Figure 4. Scatter plot of uncertainty in BFI ($W_{BFI}$) and mean uncertainty in $f_{bf}$ (Mean $W_{fbf}$).**