# Peer review of "Technical note: Analytical sensitivity analysis and uncertainty of baseflow index calculated by a two-component estimation hydrograph separation method with conductivity as a tracer"

_Hydrology and Earth System Sciences, 2018_

## Referee Comment (RC1) · Anonymous Referee #1 · 26 Oct 2018

The authors present a reasonable supplement to the conductivity two-component hydrograph separation method.

I have found only one smaller calculation error. Page 3, equation 14, and page 8, equations A8 and C1: In the second term in the numerator of the partial derivative dBFI/dyk a factor n is missing. It comes from the partial derivative of the sum of yk with respect to yk: (sum yk)' = d(sum yk)/dyk = sum dyk/dyk = n times 1 = n.

Some formulations are linguistically or technically incorrect. Please consider the following suggestions:

page 2, lines 10 - 11: Parameter sensitivity, as I understand this term, is the sensitivity of model output to the varying values of model input and not "the sensitivity of the parameters". Also, better than "fluctuation parameters" would perhaps be "varying parameter values". Replace "Eckhradt" by "Eckhardt".

page 2, line 12: Replace "An empirical sensitivity analysis is only an analytical calculation of the error propagation through the model, is not feasible." by "An empirical sensitivity analysis is only a makeshift if an analytical sensitivity analysis, that is an analytical calculation of the error propagation through the model, is not feasible".

page 2, line 14 Replace "However, the" by "Until now, the".

page 3, lines 5-6: Replace "the BFI' errors caused by tiny errors of BFC and ROC can be expressed as" by "the errors of BFI caused by small errors of BFC and ROC can be approximated by".

Throughout the paper, the sensitivity indices should be noted with vertical bars, and not with slashes (e. g. S(BFI|BFc) instead of S(BFI/BFc)).

page 3, lines 16 - 17: Replace "e.g. S(BFI/BFc) = 1.5, the relative error of BFc is 5%, then the relative error of BFI should be 1.5 times 5% (7.5%)" by "e.g. if S(BFI|BFc) = 1.5, and the relative error of BFc is 5%, then the relative error of BFI is 1.5 times 5% = 7.5%"

page 3, line 26: If the unit of Qck is $\mu$s/cm, then the unit of the partial derivative of BFI with respect to Qck is cm/$\mu$s.

page 3, line 27, page 8, lines 16 and 22: If the unit of yk is m3/d, then the unit of the partial derivative of BFI with respect to yk is d/m3.

page 4, line 3: Omit "usually".

page 4, lines 2-5, and lines 6-10: These two paragraphs express one and the same

("the error of BFI caused by the errors of Qck and yk can be neglected"). Then, this is empirically shown again in rest of this section, including figures 1 and 2. Is this necessary? If the sum of delta Qck and the sum of delta yk were not zero for n to infinity, then delta Qck and delta yk did not stand for random errors, but for systematic errors.

page 5, line 3: Replace "a parameter g is calculated" by "a variable g is calculated".

page 5, line 5: Equation 17 is the Gaussian error propagation. The citation "(Taylor, 1982; Kline, 5 1985; Genereux, 1998)" is not appropriate in this context.

page 6, line 24: Replace "The sensitivity index" by "The absolute value of the sensitivity index".

page 6, line 25: Replace "-1.39 times of uncertainty in BFI (-6.95%), while ROC leads to -0.98 times (4.9%)" by "-1.39 times 5 % of uncertainty in BFI (-6.95%), while ROC leads to -0.98 times 5 % (4.9%)".

Fig. 3: Replace "normal axes" by "linear axes".
* * *

---

## Referee Comment (RC2) · I. Cartwright (Referee) · 28 Oct 2018

The authors present an interesting analysis of sensitivity of the two-component chemical mass balance method based on specific conductivity (SC). The paper is generally clear, although the final version should be read carefully for grammar and understanding.

My main concern is how the errors in the baseflow SC are dealt with. As noted by the authors, this has a major impact on the results of the chemical mass balance. Aside

from the question as to whether to use the 99th percentile or the maximum SC, there are several common ways of estimating the SC of baseflow in chemical mass balance studies, these include:

1) Measurement in near-river groundwater bores

2) Using a single value based on the highest SC of the river throughout the study period

3) For multi-year studies, assigning a constant value for each water year (generally based on the highest SC in low summer flows)

4) Assuming that the baseflow SC varies linearly between the SC of successive low flow periods (the paper of Miller et al., 2014 uses that strategy).

These strategies can produce very different estimates of baseflow from the same river SC data. This is especially true for catchments where the contrast between the SC of surface runoff and baseflow are large and where the maximum SC in the river varies between successive low flows.

In practice, it is very difficult to estimate the SC of baseflow due to

\* Groundwater having spatially variable SC and the fluxes of groundwater from different areas of the catchment varying over time as water tables rise and fall

\* Baseflow being comprised of different components (groundwater, interflow, bank return waters), all of which have different SC, that contribute to river flow in different proportions at different times.

An uncertainty of 5% (section 4.2) is probably over optimistic. In section 4.1, it would be better to calculate an uncertainty based on the last three strategies noted above (perhaps with or without the 99th percentile constraint as well). While there is no foolproof methodology for estimating the SC of baseflow, this would yield a better estimate of what the realistic uncertainties are.

Other minor comments

Equation (1). Suggest changing the nomenclature – Q is commonly used for stream-flow in papers.

Somewhere in the introduction, you should outline the necessary conditions for chemical mass balance

a) Contributions from end-members other than baseflow and surface runoff are negligible

b) The SC of runoff and baseflow are constant (or vary in a known way) over the period of record

c) Instream processes (such as evaporation) do not change SC makedly

d) Baseflow and surface runoff have significantly different SC

Check consistency with spelling of Eckhardt throughout

Page 2 lines 12-15 is not very clearly written – try to rephrase it

Section 2.2. The errors in streamflow y are only briefly discussed. The value of 3% may be fine but this value looks to come from a thesis and it is not certain whether the gauges studied are relevant to this study. Presumably someone has addressed this for the USGS gauges? More justification of this value is needed.

Page 4, Lines 10-30. Do you need this amount of detail for these minor errors? Perhaps keep the text as is, but I do not think that the figures are strictly necessary.

Page 5, line 6. State the assumptions that the uncertainties are uncorrelated and have a Gaussian distribution.

Section 4. This application is appropriate but as noted above, the uncertainties in the SC of the baseflow (and possibly yk) are understated.

Conclusions. You should add a sentence or two stating what the main sources of error are and how practitioners can go about reducing those. For example, better rating

curves are probably more important than better loggers and more work on understanding the SC of baseflow (although it is not clear how you might do that) is more important than understanding the SC of surface runoff.

---

## Referee Comment (RC3) · Anonymous Referee #3 · 7 Nov 2018

**General comments**

The paper by Yang et al. presents a methodology to compute the uncertainty in the estimation of the long-term baseflow index (BFI) from streamflow and conductance timeseries in rivers. The paper develops equations on the sensitivity of the BFI that, to my knowledge, are new. However, I find that the overall significance of the paper is rather limited. In particular:

1) The authors mention in the title the "two-component hydrograph separation". This

is a rather broad and active field of research but the authors narrow their focus on one single index (the BFI, which expresses the long-term ratio between baseflow and streamflow) and they compute it with a very specific methodology.

2) The methodology for the hydrograph separation (equation 1) is based on several assumptions (not mentioned in the manuscript) that are typically not met in the field. One of these is the fact that the parameters of equation (1) are supposed to be fixed during an event (or for an entire timeseries, as done by the authors). Finding a methodology to relax those assumptions is, in my view, more useful than evaluating the sensitivity of the present methodology to small measurement errors.

In other words, I feel that the authors improve the uncertainty evaluation of an index that, as currently defined, has major constraints and limited reliability.

**Specific comments**

Variable names are rather confusing to a hydrologic community, as Q is conventionally used for streamflow. I invite the authors to adopt a notation based on the papers they refer to (e.g. Miller et al. 2014, Genereux 1998).

Besides English grammar errors, the language needs to be improved as the text is often difficult to understand. I invite the authors to revise the use of the term "specific": it seems that they use "specific" to say computed/available. (e.g. specific discharge appears to be just an available timeseries of discharge). Similarly, the use of "specific values" at page 3 Line 17 and "specific" conductivity values (the correct form is specific conductance or electrical conductivity)

Section 2.2. What is, ultimately, the purpose of this section? Is it to show that the sensitivity of BFI on streamflow and conductance measurements is low (and so it can be removed from subsequent equations like eq 20)?. If so, please make it clearer. What sounds interesting to me is that BFI sensitivity only depends on the integral of the (little) errors on Q and y. But once this is clear from the formula (eq. 15 and 16),

then there is no need to show Figures 1 and 2 as the result is implicit from the definition of random errors on Q and y. Instead of the current Figures 1 and 2, why not showing an example of the methodology applied to a case study time series? It would make it easier to understand the usefulness of the approach.

Section 3.1: Please make explicit assumptions on the requirement to apply the error propagation formula (eq 17). For example, "tiny" errors means that errors on Q and y should be small random errors related to the analytic uncertainty of the instrument, i.e. no systematic error.

Line 22: rather than "can effectively identify" use "aims to identify"
Line 25: "is considered the most effective separation method". By which standards?

Line 1: I guess this is limited to the particular conditions at which Stewart et al (2007) applied the method. But this is not enough to generalize.
Line 30: here and after I guess it should be equation (A1) rather than Appendix A1

Line 2: unclear what is meant by "random analysis errors". Please define what you mean by "tiny errors in Qck and yk".
Line 2-5: This statement is unjustified. Please either formulate it as a hypothesis (e.g., if the errors follow a normal distribution. . . ) or remove it.
Lines 6-7: "The uncertainty of [. . .] is. . .": please avoid these unjustified general statements. Instrument precision depends on the particular instrument at hand and streamflow precision depends on a very large number of factors. You can simply reformulate the sentence stating that you assumed errors of 5Lines 11-18 is particularly unclear
Line 17: which "average error (

Line 13-17: what is the rationale behind the choice of these particular types of uncertainty (W terms)?

Line 4-5: "During the rainstorm [. . .] the streamflow is almost entirely from the rainfall runoff": this is a serious misinterpretation of hydrological processes. It is well known since at least 15 years that in most catchments the event-water is not a major component of streamflow (and very often it only accounts for a few percent of total flow). See e.g. the commentary by Kirchner (2003) on Hydrological Processes (https://doi.org/10.1002/hyp.5108).

––––––––––––––––––––––––––––––––––––

---

## Author Comment (AC1) · 26 Nov 2018

**Response to anonymous Reviewer #1**

**Reply explanation:** The reviewers' comments are shown in black, while the author's replies and revises are shown in blue.

**Comments:** The authors present a reasonable supplement to the conductivity two-component hydrograph separation method.

**Reply:** We appreciate the positive comment for this study.

**Comments:** I have found only one smaller calculation error. Page 3, equation 14, and page 8, equations A8 and C1: In the second term in the numerator of the partial derivative dBFI/dyk a factor n is missing. It comes from the partial derivative of the sum of yk with respect to yk: (sum yk)' = d(sum yk)/dyk = sum dyk/dyk = n times 1 = n.

**Reply and Revise:** Well taken, thank you very much for finding out our errors. We have rechecked all the equations in the manuscript and revised the errors (Page 4, equations 12, 14 and16; Page 9, equations A8; Page 10 equation C1, and line 10).

**Comments:** Some formulations are linguistically or technically incorrect. Please consider the following suggestions: page 2, lines 10 - 11: Parameter sensitivity, as I understand this term, is the sensitivity of model output to the varying values of model input and not "the sensitivity of the parameters". Also, better than "fluctuation parameters" would perhaps be "varying parameter values". Replace "Eckhradt" by "Eckhardt".

**Reply and Revise:** Well taken, we also understand that parameter sensitivity is the sensitivity of model output to the varying values of model input. There may be some errors in the expression of the manuscript, and we have revised it (Page 2, lines 24--25). And we have replaced "fluctuation parameters" by "varying parameter values" (Page 2, lines 24--25), also "Eckhradt" by "Eckhardt" (Page 2, line 26).

**Comments:** page 2, line 12: Replace "An empirical sensitivity analysis is only an analytical calculation of the error propagation through the model, is not feasible." by "An empirical sensitivity analysis is only a makeshift if an analytical sensitivity analysis, that is an analytical calculation of the error propagation through the model, is not feasible".

**Reply and Revise:** Well taken, we have replaced the sentence as suggested (Page 2, lines 26--28).

**Comments:** page 2, line 14: Replace "However, the" by "Until now, the".

**Reply and Revise:** Well taken, we have replaced the words as suggested (Page 2, line 30).

**Comments:** page 3, lines 5-6: Replace "the BFI' errors caused by tiny errors of $BF_C$ and $RO_C$ can be expressed as" by "the errors of BFI caused by small errors of $BF_C$ and $RO_C$ can be approximated by".

**Reply and Revise:** Well taken, we have replaced the sentence as suggested (Page 3, lines 24--25).

**Comments:** Throughout the paper, the sensitivity indices should be noted with vertical bars, and not with slashes (e. g. S(BFI|BFc) instead of S(BFI/BFc))

**Reply and Revise:** Well taken, we have revised the sensitivity indices throughout the paper as suggested.

**Comments:** page 3, lines 16 - 17: Replace "e.g. S(BFI/BFc) = 1.5, the relative error of BFc is 5%, then the relative error of BFI should be 1.5 times 5% (7.5%)" by "e.g. if S(BFI|BFc) = 1.5, and the relative error of BFc is 5%, then the relative error of BFI is 1.5 times 5% = 7.5%"

**Reply and Revise:** Well taken, we have replaced the sentence as suggested (Page 4, lines 4--5).

**Comments:** page 3, line 26: If the unit of Qck is $\mu s$/cm, then the unit of the partial derivative of BFI with respect to Qck is cm/$\mu s$.

**Reply and Revise:** Well taken, we have added the unit of the partial derivative of BFI with respect to $Q_{ck}$ (Page 4, line 17).

**Comments:** page 3, line 27, page 8, lines 16 and 22: If the unit of yk is m3/d, then the unit of the partial derivative of BFI with respect to yk is d/m3.

**Reply and Revise:** Well taken, we have added the unit of the partial derivative of BFI with respect to $y_k$ (Page 4, line 17; Page 10, lines 8 and 14).

**Comments:** page 4, line 3: Omit "usually".

**Reply and Revise:** Well taken, we have deleted the "usually".

**Comments:** page 4, lines 2-5, and lines 6-10: These two paragraphs express one and the same ("the error of BFI caused by the errors of Qck and yk can be neglected"). Then, this is empirically shown again in rest of this section, including figures 1 and 2. Is this necessary? If the sum of delta Qck and the sum of delta yk were not zero for n to infinity, then delta Qck and delta yk did not stand for random errors, but for systematic errors.

**Reply and Revise:** Well taken, we have reduced the description of this section and have added Fig. 1 and Fig. 2 and related descriptions to the Supplement S1 (Page 5, lines 3--31).

**Comments:** page 5, line 3: Replace "a parameter g is calculated" by "a variable g is calculated".

**Reply and Revise:** Well taken, we have replaced the words as suggested (Page 5, line 34).

**Comments:** page 5, line 5: Equation 17 is the Gaussian error propagation. The citation "(Taylor, 1982; Kline, 5 1985; Genereux, 1998)" is not appropriate in this context.

**Reply and Revise:** Well taken, we have adjusted the description and citation of equation 17 (Page 5, lines 34--37).

**Comments:** page 6, line 24: Replace "The sensitivity index" by "The absolute value of the sensitivity index".

**Reply and Revise:** Well taken, we have replaced the words as suggested (Page 7, line 31).

**Comments:** page 6, line 25: Replace "-1.39 times of uncertainty in BFI (-6.95%), while $RO_C$ leads to -0.98 times (4.9%)" by "-1.39 times 5 % of uncertainty in BFI (-6.95%), while $RO_C$ leads to -0.98 times 5 % (4.9%)".

**Reply and Revise:** Well taken, we have replaced the sentence as suggested (Page 7, lines 33--34).

**Comments:** Fig. 3: Replace "normal axes" by "linear axes".

**Reply and Revise:** Well taken, we have replaced the words as suggested (Fig. 1)

**Language Improve:** We have asked an English native language agency to check and correct the grammar and structure of the manuscript.

---

## Author Comment (AC2) · 26 Nov 2018

**Response to anonymous Reviewer #3**

**Reply explanation:** The reviewers' comments are shown in black, while the author's replies and revises are shown in blue.

**General comments:**

**Comments:** The paper by Yang et al. presents a methodology to compute the uncertainty in the estimation of the long-term baseflow index (BFI) from streamflow and conductance timeseries in rivers. The paper develops equations on the sensitivity of the BFI that, to my knowledge, are new. However, I find that the overall significance of the paper is rather limited. In particular:

1) The authors mention in the title the "two-component hydrograph separation". This is a rather broad and active field of research but the authors narrow their focus on one single index (the BFI, which expresses the long-term ratio between baseflow and streamflow) and they compute it with a very specific methodology.

2) The methodology for the hydrograph separation (equation 1) is based on several assumptions (not mentioned in the manuscript) that are typically not met in the field. One of these is the fact that the parameters of equation (1) are supposed to be fixed during an event (or for an entire time series, as done by the authors). Finding a methodology to relax those assumptions is, in my view, more useful than evaluating the sensitivity of the present methodology to small measurement errors.

In other words, I feel that the authors improve the uncertainty evaluation of an index that, as currently defined, has major constraints and limited reliability.

**Reply:** We are very grateful to the anonymous reviewer for reviewing the paper and affirming the sensitivity equation. There are two main purposes in this paper. One is to analyze the sensitivity of the long-term series of baseflow separation results (BFI) to the parameters and variables in the conductivity two-component hydrograph separation equation (Sect. 2), and the other is to derive the uncertainty of BFI (Sect.3).(Page 3, lines 1--3).

Sensitivity analysis can quantitatively describe the impact of parameters and variables on the results of base flow separation (so that future users can clearly know which parameter has a greater impact, and should be more cautious when choosing the value). Uncertainty analysis can quantitatively describe the trusted range of BFI calculated by the conductivity two-component hydrograph separation method. The sensitivity analysis equation and uncertainty analysis equation of this paper can be easily applied to the two-component baseflow separation methods of other tracers because they usually have a unified equation form.

**Reply to 1):** BFI is a form of presentation of long-term series of baseflow separation results. And BFI is a hydrogeological parameter useful in modeling un-gaged basins (Lott and Stewart, 2016) and is believed to represent the effect of geology on basin low flows (Gustard et al., 1992). The total amount of baseflow in a long-term series can be easily obtained by multiplying the BFI by the total streamflow, where the long-term series can be months or years. Researchers in water resources management and assessment usually want to analyze the transformation between groundwater and streamflow by determining the baseflow under a long-term series, so this paper is necessary for them.

Stewart et al. (2007) applied the conductivity two-component hydrograph separation method (also known as CMB) to 10 real-time USGS gauging stations in Florida, Georgia, Texas, and Kentucky. Cartwright et al. (2014) applied this method to the Barwon River catchment, southeast Australia. Miller et al. (2014) applied this method to estimate the baseflow in 14 snowmelt-dominated streams and rivers in the Upper Colorado River Basin. Mei and Anagnostou (2015) indicated that the tracer based hydrograph separation method yields the most realistic results among various methods, because the tracer based method with the highest physical basis. Lott and Stewart (2016) applied this method to 35 basins in USA to calibrate five other baseflow separation methods. The conductivity two-component hydrograph separation method has its own limitations, but there seems to be no better method at present (Miller et al., 2014; Lott and Stewart, 2016).

**Reply and Revise to 2):** The conductivity two-component hydrograph separation method is indeed based on some assumptions. We have added these assumptions to the manuscript (Page 2, lines 3--7).

The field test (which was located within a $12km^2$ drainage basin in southeast Hillsborough County, Florida.) of Stewart et al. (2007) showed that the maximum conductivity of streamflow can be used to replace $BF_C$, and the minimum conductivity can be used to replace $RO_C$. The field test may be limited, but the conclusions have been applied to many basins in USA (as mentioned above). Miller et al. (2014) pointed out that the maximum conductivity of streamflow may exceed the real $BF_C$, so they suggested that the 99th percentile of the conductivity of each year should be used as the $BF_C$ and assumed that the baseflow conductivity varies linearly between years.

Considering that the parameters of the conductivity two-component hydrograph separation method may be varied during the time series, we have changed the application of sensitivity analysis and uncertainty estimation based on the strategy of Miller et al. (2014) (Sect. 4.1 & 4.2).

**Specific comments:**

**Comments:** Variable names are rather confusing to a hydrologic community, as Q is conventionally used for streamflow. I invite the authors to adopt a notation based on the papers they refer to (e.g. Miller et al. 2014, Genereux 1998).

**Reply and Revise:** Well taken, we have changed the nomenclature and used SC as the variable name of specific conductance throughout the paper.

**Comments:** Besides English grammar errors, the language needs to be improved as the text is often difficult to understand. I invite the authors to revise the use of the term "specific": it seems that they use "specific" to say computed/available. (e.g. specific discharge appears to be just an available timeseries of discharge). Similarly, the use of "specific values" at page 3 Line 17 and "specific" conductivity values (the correct form is specific conductance or electrical conductivity).

**Reply and Revise:** Well taken, we have asked an English native language agency to check and correct the grammar and structure of the manuscript. We have revised the mistake use of the term "specific" throughout the paper.

**Comments:** Section 2.2. What is, ultimately, the purpose of this section? Is it to show that the sensitivity of BFI on streamflow and conductance measurements is low (and so it can be removed from subsequent equations like eq 20)?. If so, please make it clearer. What sounds interesting to me is that BFI sensitivity only depends on the integral of the (little) errors on Q and y. But once this is clear from the formula (eq. 15 and 16), then there is no need to show Figures 1 and 2 as the result is implicit from the definition of random errors on Q and y. Instead of the current Figures 1 and 2, why not showing an example of the methodology applied to a case study time series? It would make it easier to understand the usefulness of the approach.

**Reply and Revise:** Well taken, the ultimately purpose of Section 2.2 is indeed to show that the sensitivity of BFI on streamflow and specific conductance measurements is low (which can be ignored in estimating the uncertainty of BFI). We have made it clearer (Page 5, lines 3--6).

This section really does not need so much description. We have reduced the description and have added Fig. 1 and Fig. 2 and related descriptions to the Supplement S1 (Page 5, lines 3--31).

**Comments:** Section 3.1: Please make explicit assumptions on the requirement to apply the error propagation formula (eq 17). For example, "tiny" errors means that errors on Q and y should be small random errors related to the analytic uncertainty of the instrument, i.e. no systematic error.

**Reply and Revise:** Well taken, we have added the assumptions of the error propagation formula (Gaussian error propagation) (Page 5, lines 34--37).

**Comments:** Page 1 Line 22: rather than "can effectively identify" use "aims to identify"

**Reply and Revise:** Well taken, we have replaced the words as suggested (Page 1, line 26).

**Comments:** Page 1 Line 25: "is considered the most effective separation method". By which standards?

**Reply and Revise:** *Kendall* et al. (1998) and Miller et al. (2014) indicated that stable isotopes are generally considered to be the most accurate chemical tracers for hydrograph separation. Klaus et al. (2013) and Lott and Stewart *(2016)* indicated that stable isotope tracers are considered to be the best geochemical method for hydrograph separation. Mei and Anagnostou (2015) indicated that the tracer based hydrograph separation method yields the most realistic results among various methods, because the tracer based method with the highest physical basis.

This paper follows the statement in the above articles. Our statement may not be objective enough, so we have made some changes (Page 1, lines 29--32).

**Comments:** Page 2 Line 1: I guess this is limited to the particular conditions at which Stewart et al (2007) applied the method. But this is not enough to generalize.

**Reply and Revise:** The field test site of Stewart et al. (2007) was located within a 12km$^2$ drainage basin in southeast Hillsborough County, Florida. The conclusions (the maximum conductivity of streamflow can be used to replace $BF_C$, and the minimum conductivity can be used to replace $RO_C$) of the test were applied to 10 real-time USGS gauging stations in Florida, Georgia, Texas, and Kentucky. Then, the conclusions were applied to 35 basins in USA to calibrate five other baseflow separation methods (Lott and Stewart, *2016*).

The field test may be limited, but the conclusions have been applied to many basins in USA. Considering these, we have added the particular conditions for the field test to the manuscript (Page 2, line 13).

**Comments:** Page 2 Line 30: here and after I guess it should be equation (A1) rather than Appendix A1

**Reply and Revise:** Well taken, we have replaced the presentation of the citations as suggested (Page 3, line 10; Page 3, lines 20 and 30; Page 4, line 10).

**Comments:** Page 4 Line 2: unclear what is meant by "random analysis errors". Please define what you mean by "tiny errors in Qck and yk".

**Reply and Revise:** Well taken, we have revised the description of this paragraph (Page 4, lines 27--35).

**Comments:** Page 4 Line 2-5: This statement is unjustified. Please either formulate it as a hypothesis (e.g., if the errors follow a normal distribution: : : ) or remove it.

**Reply and Revise:** Well taken, we have removed it.

**Comments:** Page 4 Lines 6-7: "The uncertainty of [: : :] is: : :": please avoid these unjustified general statements. Instrument precision depends on the particular instrument at hand and streamflow precision depends on a very large number of factors. You can simply reformulate the sentence stating that you assumed errors of Lines 11-18 is particularly unclear

**Reply and Revise:** Well taken, we have changed the statements.

**Comments:** Page 4 Line 17: which "average error"

**Reply and Revise:** We have moved this paragraph to the Supplement S1. It should be "relative error (%) of $\sum_{k=1}^{n} SC_k$ and $\sum_{k=1}^{n} y_k$" not "average error", and we have revised it.

**Comments:** Page 5 Line 13-17: what is the rationale behind the choice of these particular types of uncertainty (W terms)?

**Reply and Revise:** The uncertainty terms in Gaussian error propagation should be of the same type. One has some choice in the type of uncertainty to propagate, but all the uncertainty values must be the same kind of quantity: either all average errors, all standard deviations, etc. (Genereux, 1998; Ernest, 2005).

"While any set of consistent uncertainty (W) values may be propagated using Gaussian error propagation, using standard deviations multiplied by $t$ values from the Student's $t$ distribution (each $t$ for the same confidence level, such as 95%) has the advantage of providing a clear meaning (tied to a confidence interval) for the computed uncertainty would correspond to, for example, 95% confidence limits on BFI" (Genereux, 1998).

We have added this rationale to the manuscript (Page 6, lines 5--8).

**Comments:** Page 7 Line 4-5: "During the rainstorm [: : :] the streamflow is almost entirely from the rainfall runoff": this is a serious misinterpretation of hydrological processes. It is well known since at least 15 years that in most catchments the event-water is not a major component of streamflow (and very often it only accounts for a few percent of total flow). See e.g. the commentary by Kirchner (2003) on Hydrological Processes (https://doi.org/10.1002/hyp.5108).

**Reply and Revise:** We are very grateful to the reviewer for pointing out this problem and giving the reference. Previously, we were mainly concerned about the effect of water-rock interaction on the chemical composition of groundwater and baseflow, and missed the brilliant debate on the "old water paradox".

After reading Kirchner's (2003) article, we thought the "old water paradox" is that in the same basin, most isotope tracers show that the flood streamflow contains a large amount of pre-event water, while dissolved salt tracers show that the flood streamflow contains less pre-event water (Isotopic fluctuations are very small in the process of flood streamflow, while the fluctuation of dissolved salts is more obvious).

Van Verseveld (2009) tried to explain the "old water paradox" through a hillside sprinkling experiment, the results show that "mass transfer to the immobile domain, dispersive mixing and rapid transport via lateral subsurface flow explained rapid mobilization of old water and thus the first part of the double paradox in a plausible mechanistic way", "the supply limitation of DOC, in combination with the vertical and lateral flow paths, controlled the variable DOC chemistry in lateral subsurface flow". Kienzler (2010) also tried to explain the first part of the "old water paradox" through some hillside sprinkling experiments, the results show that "shallow soil may already contain significant amounts of pre-event water, which can be rapidly released from small, saturated patches of the soil matrix", "an intensive exchange between overland flow and shallow subsurface flow might be quite common, … overland flow and fast subsurface flow, may, at the same time, produce rapid discharge responses and deliver substantial amounts of pre-event water to the stream". We have not found a strong theory to solve this paradox, and the existing theories are mostly plausible.

Just our opinion, the isotope composition of rainfall runoff has changed significantly in the surface or shallow soil, while the change in dissolved salt is not obvious. Therefore, the isotope tracer may classify the soil flow and the return flow into the subsurface runoff, while the dissolved salt tracer may classify the soil flow and the return flow into the surface runoff. And we do not think that soil flow and return flow are strictly subsurface runoff (which should be the water flow through the aquifer with uniform hydraulic connection). The above is just our experience. We believe that it is difficult to say that event-water is not a major component of streamflow until this paradox is thoroughly explained or proven to be correct in one part and biased in the other.

Considering the existence of the "old water paradox", we have removed the description that may be wrong in the paper (Page 8, lines 14--16).

**References:**

*Stewart, M., Cimino, J., and Rorr, M.: Calibration of base flow separation methods with streamflow conductivity, Ground Water, 45, 17-27, doi:10.1111/j.1745-6584.2006.00263.x, 2007.*

*Lott, D. A., & Stewart, M. T. . (2016). Base flow separation: a comparison of analytical and mass balance methods. Journal of Hydrology, 535, 525-533.*

*Gustard, A., Bullock, A., Dixon, J.M., 1992. Low Flow Estimation in the United Kingdom. Institute of Hydrology, Wallingford, p. 88 (IH Report No.108).*

Cartwright, I., Gilfedder, B., and Hofmann, H.: Contrasts between estimates of baseflow help discern multiple sources of water contributing to rivers, Hydrol. Earth Syst. Sci., 18, 15-30, doi:10.5194/hess-18-15-2014, 2014.

*Kendall, S., and E. A. Caldwell (1998), Fundamentals of isotope geochemistry, in Isotope Tracers In Catchment Hydrology, edited by C. Kendall and J. J. McDonnell, pp. 51–86, Elsevier, Amsterdam.*

*Miller, M. P., Susong, D. D., Shope, C. L., Heilweil, V. M., and Stolp, B. J.: Continuous estimation of baseflow in snowmelt-dominated streams and rivers in the Upper Colorado River Basin: A chemical hydrograph separation approach, Water Resour. Res., 50, 6986–6999, doi:10.1002/2013WR014939, 2014.*

Klaus, J., McDonnell, J.J., 2013. Hydrograph separation using stable isotopes: review and evaluation. J. Hydrol. 505, 47–64. http://dx.doi.org/10.1016/j.jhydrol.2013. 09.006.

Lott D A , Stewart M T . Base flow separation: A comparison of analytical and mass balance methods[J]. Journal of Hydrology, 2016, 535:525-533.

Mei, Y. , & Anagnostou, E. N. . (2015). A hydrograph separation method based on information from rainfall and runoff records. Journal of Hydrology, 523, 636-649.

Genereux, D.: Quantifying uncertainty in tracer-based hydrograph separations, *Water Resour. Res, 34, 915-919, doi:10.1029/98wr00010, 1998.*

Ernest, L.: Gaussian error propagation applied to ecological data: Post-ice-storm-downed woody biomass, Ecol. Monogr., 75, 451-466, doi.org/10.1890/05-0030, 2005.

*Kienzler, P. M. . (2010). Subsurface storm flow formation at different hillslopes and implications for the 'old water paradox'. Hydrological Processes, 22(1), 104-116.*

Van Verseveld, W. J , et al. "A hillslope scale sprinkling experiment to resolve the double paradox in hydrology." Egu General Assembly Conference EGU General Assembly Conference Abstracts, 2009.

---

## Author Comment (AC3) · 26 Nov 2018

**Reply explanation:** The reviewers' comments are shown in black, while the author's replies and revises are shown in blue.

**Comments:** The authors present an interesting analysis of sensitivity of the two-component chemical mass balance method based on specific conductivity (SC). The paper is generally clear, although the final version should be read carefully for grammar and understanding.

**Reply and Revise:** We appreciate the positive comment for this study. And we have asked an English native language agency to check and correct the grammar and structure of the manuscript.

**Comments:** My main concern is how the errors in the baseflow SC are dealt with. As noted by the authors, this has a major impact on the results of the chemical mass balance. Aside from the question as to whether to use the 99th percentile or the maximum SC, there are several common ways of estimating the SC of baseflow in chemical mass balance studies, these include:

1) Measurement in near-river groundwater bores

2) Using a single value based on the highest SC of the river throughout the study period

3) For multi-year studies, assigning a constant value for each water year (generally based on the highest SC in low summer flows)

4) Assuming that the baseflow SC varies linearly between the SC of successive low flow periods (the paper of Miller et al., 2014 uses that strategy).

These strategies can produce very different estimates of baseflow from the same river SC data. This is especially true for catchments where the contrast between the SC of surface runoff and baseflow are large and where the maximum SC in the river varies between successive low flows.

In practice, it is very difficult to estimate the SC of baseflow due to

* Groundwater having spatially variable SC and the fluxes of groundwater from different areas of the catchment varying over time as water tables rise and fall

\* Baseflow being comprised of different components (groundwater, interflow, bank return waters), all of which have different SC, that contribute to river flow in different proportions at different times.

An uncertainty of 5% (section 4.2) is probably over optimistic. In section 4.1, it would be better to calculate an uncertainty based on the last three strategies noted above (perhaps with or without the 99th percentile constraint as well). While there is no foolproof methodology for estimating the SC of baseflow, this would yield a better estimate of what the realistic uncertainties are.

**Reply and Revise:** We are very grateful for your explanation of the common strategies of estimating the conductivity of baseflow and the complexity of the conductivity of baseflow. We fully agree with your description.

We have recalculated the sensitivity indices and the uncertainty based on the fourth strategy (the baseflow conductivity varies linearly between the conductivity of successive low flow periods) you mentioned with the $99^{th}$ percentile constraint (Sect. 4.1; Sect. 4.2; Table 1; Figures 1&2).

Based on the results of the recalculation, we found that the mean uncertainty of $BF_C$ is about 10%, and the original use of 5% is indeed too optimistic. We have changed the uncertainty (Page 7, lines 32--33).

**Other minor comments:**

**Comments:** Equation (1). Suggest changing the nomenclature – Q is commonly used for streamflow in papers.

**Reply and Revise:** Well taken, we have changed the nomenclature and used SC as the variable name of specific conductance throughout the paper.

**Comments:** Somewhere in the introduction, you should outline the necessary conditions for chemical mass balance

a) Contributions from end-members other than baseflow and surface runoff are negligible

b) The SC of runoff and baseflow are constant (or vary in a known way) over the period of record

c) Instream processes (such as evaporation) do not change SC makedly

d) Baseflow and surface runoff have significantly different SC

**Reply and Revise:** Well taken, we have added the assumptions of conductivity two-component hydrograph separation method (chemical mass balance method) as suggested (Page 2, lines 3--7).

**Comments:** Check consistency with spelling of Eckhardt throughout

**Reply and Revise:** Well taken, we have checked the spelling and revised the mistake (Page 2, line 26).

**Comments:** Page 2 lines 12-15 is not very clearly written – try to rephrase it

**Reply and Revise:** Well taken, we have rephrased the sentences (Page 2, lines 26--28).

**Comments:** Section 2.2. The errors in streamflow y are only briefly discussed. The value of 3% may be fine but this value looks to come from a thesis and it is not certain whether the gauges studied are relevant to this study. Presumably someone has addressed this for the USGS gauges? More justification of this value is needed.

**Reply and Revise:** Well taken, the value of 3% came from the uncertainty analysis in streamflow of Yellow River, China. This value may be unfair for the USGS gauges, so we have read some articles to find a more reasonable value.

Olson et al. (2007) and Sauer et al. (2010) indicated that the continuous records of water level in USGS gauges are translated to streamflow by applying the rating curve. And the water level measurements are accurate to the nearest 0.01 foot or 0.2 percent of water level. However, we did not find a description of the uncertainty in streamflow on the website of USGS. Hamilton et al. (2012) indicated that streamflow data from USGS are often assumed by analysts to be accurate and precise to within $\pm 5\%$ at the 95% confidence interval.

Based on the above, we have revised the value of 3% to 5%, and have revised the references and related content (Page 4, lines 32--33).

**Comments:** Page 4, Lines 10-30. Do you need this amount of detail for these minor errors? Perhaps keep the text as is, but I do not think that the figures are strictly necessary.

**Reply and Revise:** Well taken, we have reduced the description of this section and have added Fig. 1 and Fig. 2 and related descriptions to the Supplement S1 (Page 5, lines 3--31).

**Comments:** Page 5, line 6. State the assumptions that the uncertainties are uncorrelated and have a Gaussian distribution.

**Reply and Revise:** Well taken, we have added the assumptions as suggested (Page 5, line 35).

**Comments:** Section 4. This application is appropriate but as noted above, the uncertainties in the SC of the baseflow (and possibly yk) are understated.

**Reply and Revise:** Well taken, we have recalculated the sensitivity indices and the uncertainty based on the fourth strategy (the baseflow conductivity varies linearly between the conductivity of successive low flow periods) you mentioned above with the 99$^{th}$ percentile constraint (Sect. 4.1; Sect. 4.2; Table 1; Figures 1&2).

Based on the results of the recalculation, we found that the mean uncertainty of $BF_C$ is about 10%, and the original use of 5% is indeed too optimistic. We have changed the uncertainty (Page 7, lines 32-33).

**Comments:** Conclusions. You should add a sentence or two stating what the main sources of error are and how practitioners can go about reducing those. For example, better rating curves are probably more important than better loggers and more work on understanding the SC of baseflow (although it is not clear how you might do that) is more important than understanding the SC of surface runoff.

**Reply and Revise:** Well taken, we have added the relevant content as suggested (Page 9, lines 6--9).

**References:**

Olson, S.A., and Norris, J.M., 2007, U.S. Geological Survey streamgaging…from the National Streamflow Information Program: U.S. Geological Survey Fact Sheet 2005–3131, 4 p. (Also available at http://pubs.usgs.gov/fs/2005/3131/.)

Sauer, V.B., and Turnipseed, D.P., 2010, Stage measurement at gaging stations: U.S. Geological Survey Techniques and Methods, book 3, chap. A7, 45 p. (Also available at http://pubs.usgs.gov/tm/tm3-a7/.)

Hamilton, A.S., and Moore R.D., 2012, Quantifying Uncertainty in Streamflow Records , Canadian Water Resources Journal / Revue canadienne des ressources hydriques, 37:1, 3-21, DOI: 10.4296/cwrj3701865

---

## Author Comment (AC4) · 26 Nov 2018

We have revised the manuscript based on all comments and attached the marked revised manuscript as a supplement.

Please also note the supplement to this comment:
https://www.hydrol-earth-syst-sci-discuss.net/hess-2018-492/hess-2018-492-AC4-supplement.pdf

---

## Author Comment (AC5) · 26 Nov 2018

[revised manuscript text omitted]
 BFI}{\partial BF_c} = \frac{y RO_c - \sum_{k=1}^{n} y_k SCQ_{ek}}{y(BF_c - RO_c)^2} \tag{5}$$

$$\frac{\partial BFI}{\partial RO_c} = \frac{\sum_{k=1}^{n} y_k SCQ_{ek} - y BF_c}{y(BF_c - RO_c)^2} \tag{6}$$

 The definition of the partial derivative suggests that the influence of the errors of the parameters ($\Delta BF_C$ and $\Delta RO_C$) in Eq. (1) on the BFI can be expressed by the product of the errors and its partial derivatives. Then the errors of BFI caused by small errors of $BF_C$ and $RO_C$ can be approximated by :

$$\Delta_{BF_c} BFI = \frac{\partial BFI}{\partial BF_c} \Delta BF_c = \frac{y RO_c - \sum_{k=1}^{n} y_k SCQ_{ek}}{y(BF_c - RO_c)^2} \Delta BF_c \tag{7}$$

$$\Delta_{RO_c} BFI = \frac{\partial BFI}{\partial RO_c} \Delta RO_c = \frac{\sum_{k=1}^{n} y_k SCQ_{ek} - y BF_c}{y(BF_c - RO_c)^2} \Delta RO_c \tag{8}$$

The dimensionless sensitivity indices (S) can be obtained by comparing the relative error of BFI caused by the small errors of $BF_C$ and $RO_C$ with that of $BF_C$ and $RO_C$, (see  Eq. (B1), (B2)):

$$S(BFI|BF_c \text{}) = \frac{\Delta_{BF_c} BFI}{BFI} \bigg/ \frac{\Delta BF_c}{BF_c} = \frac{BF_c(y RO_c - \sum_{k=1}^{n} y_k SCQ_{ek})}{y BFI(BF_c - RO_c)^2}$$
$$\tag{9}$$

$$S(BFI|RO_c \text{}) = \frac{\Delta_{RO_c} BFI}{BFI} \bigg/ \frac{\Delta RO_c}{RO_c} = \frac{RO_c(\sum_{k=1}^{n} y_k SCQ_{ek} - y BF_c)}{y BFI(BF_c - RO_c)^2}$$
$$\tag{10}$$

where $S(BFI|BF_c$ $)$ represent the dimensionless sensitivity index of BFI (output) with $BF_c$ (uncertain input), and $S(BFI|RO_c$ $)$ with $RO_c$.

The dimensionless sensitivity index is also called the "elasticity index", and it reflects the proportional relationship between the relative error of BFI and the relative error of parameters (e.g. if $S(BFI|BF_c$ $) = 1.5$, and the relative error of $BF_c$ is 5%, then the relative error of BFI is 1.5 times 5%  7.5% ). After determining the  values of $BF_C$, $RO_C$, BFI, $y$, $y_k$ and $SC_{ek}$, the sensitivity indices $S(BFI|BF_c$ $)$ and $S(BFI|RO_c$ $)$ can be calculated and compared.

**2.2 Variables $y_k$ and $SC_{ek}$**

In addition to the two parameters, there are two variables ($SC_{ek}$ and $y_k$) in Eq. (1). This section describes the sensitivity analysis of BFI to these two variables. Similar to Sect. 2.1, the partial derivatives of $b_k$ in Eq. (1) to $SC_{ek}$ and $y_k$ are obtained (see Eq. (A5), (A6)), and the partial derivatives of BFI to $SC_{ek}$ and $y_k$ are further obtained (see Eq. (A7), (A8)),

$$\frac{\partial BFI}{\partial sc_{ek}} = \frac{1}{BF_c - RO_c} \tag{11}$$

$$\frac{\partial BFI}{\partial y_k} = \frac{\sum_{k=1}^{n}(sc_{ek}-RO_c)-nBFI(BF_c-RO_c)}{y(BF_c-RO_c)} \tag{12}$$

According to previous studies (Munyaneza et al., 2012; Cartwright et al., 2014; Miller et al., 2014; Okello et al., 2018) and this study (Table 1), the difference between $BF_C$ and $RO_C$ is often greater than 100 μs/cm. Therefore,  $\partial BFI/\partial SC_{ek}$ is usually less than 0.01 cm/μs. Appendix C shows that the value of $\partial BFI/\partial y_k$ is usually far less than 1 d/m³.

Small errors in $SC_{ek}$ and $y_k$ cause errors in BFI

$$\Delta_{SC_{ek}}BFI = \frac{\partial BFI}{\partial sc_{ek}}\Delta SC_{ek} = \frac{\Delta sc_{ek}}{BF_c - RO_c} \tag{13}$$

$$\Delta_{y_k}BFI = \frac{\partial BFI}{\partial y_k}\Delta y_k = \frac{\sum_{k=1}^{n}(sc_{ek}-RO_c)-nBFI(BF_c-RO_c)}{y(BF_c-RO_c)}\Delta y_k \tag{14}$$

The errors of BFI caused by $SC_{ek}$ and $y_k$ are summed up to obtain the error of BFI caused by $\sum_{k=1}^{n}SC_{ek}$ and $\sum_{k=1}^{n}y_k$ in the whole time series:

$$\Delta_{\sum_{k=1}^{n}sc_{ek}}BFI = \sum_{k=1}^{n}\Delta_{sc_{ek}}BFI = \sum_{k=1}^{n}\frac{\Delta sc_{ek}}{
[revised manuscript text omitted]

---

## Author Response (AR2)

**Response**

*Reply explanation*: *The comments are shown in black, and the author's replies and revises are shown in blue.*

**Response to Editor**

**Comments:** The two reviewers agree that the manuscript has been substantially improved. However, one referee points out that one of his major previous comments, a discussion relating to the assumptions in equation 1, was not adequately addressed. Before this manuscript can be considered for publication, it will be necessary to include some quantification (e.g. some sort of synthetic sensitivity analysis) and a detailed discussion about potential effects of non-constant soil water conductivity on your results/interpretations.

**Reply:**

Dear editor,

Thank you very much for your handling and comments on our manuscript. We have revised the manuscript in the light of your comments and those of the reviewers. The following is our response to each comment.

Non-constant soil flow conductivity does affect our results, and we have added a corresponding discussion in the manuscript. Because we have only found few studies on the conductivity of soil flow, so we mainly discuss it qualitatively.

**Revise:** We have added the following paragraph to the manuscript (Page 7, Lines 3--12).

"The conductivity of shallow subsurface and soil flow in real watersheds is sensitive to climatic conditions and usually shows obvious fluctuations (Miller et al., 2014). The CMB method classifies high conductivity flow (e.g., deep subsurface flow) as baseflow and low conductivity flow (e.g., local shallow soil flow) as surface runoff (Cartwright et al., 2014). Therefore, in the watershed containing a large number of low-conductivity soil flows, the BFI calculated by the CMB method comprised only the baseflow index of the deep subsurface flow. The parameter sensitivity indices and uncertainty of the deep subsurface flow were also calculated by the methods of this paper. Cartwright et al. (2014) showed that the ratio of low-conductivity soil flow to high-conductivity subsurface flow in the Barwon basin in southeastern Australia is close to 1.

If only the BFI doubles and other parameters remain unchanged, then the sensitivity indices calculated by Eq. (9) and Eq. (10) are halved, whereas the uncertainty calculated by Eq. (23) remains unchanged. Therefore, non-constant soil flow conductivity may lead to an overestimation of sensitivity, but it has less impact on uncertainty estimates."

**Response to anonymous Reviewer #2**

**Comments:** This revised paper has addressed most of the major concerns of the reviewers. It is a useful addition to the field and suitable for publication as a technical note.

**Reply:** We are very grateful for your affirmation of the revised manuscript.

**Response to anonymous Reviewer #3**

**General comments:** The technical note by Weifei Yang and coauthors has improved with respect to previous version. I am still unsure that, by considering the uncertainty due to random measurement errors only, the authors are proceeding in the right direction, but as the paper can already be useful to some part of the scientific community I am glad to recommend it for publication subject to minor revisions. Please find below a note of general character (that does not need to be addresses in the revision) and then a few minor points that could help improve the final manuscript.

Assumptions behind the two-component hydrograph separation method: I think the crucial point was not to add a list of the assumptions underlying the method, but to discuss what might happen when the assumptions are not met. Equation (1) might not be good for baseflow separation, especially when soil water has a major contribution to streamflow and its conductivity is not constant over time. Then, the assumed conductivity of baseflow and surface runoff (parameters $BF_c$ and $RO_c$) are to some extent arbitrary: the authors show from the beginning (Page 2) that different choices can be made to characterize those parameters and the parameters may also vary at event or seasonal scale. All these sources of uncertainty are potentially larger than the simple random measurement errors considered in the manuscript.

**Reply:** We are very grateful for your affirmation of the revised manuscript. The non-constant soil flow conductivity does affect the results of the two-component baseflow separation method as

well as the sensitivity analysis and uncertainty estimates. We have added some corresponding discussion in the manuscript according to your and editor's comments (Page 7, Lines 3--12).

**Minor points**

**Comments:** As already mentioned in my previous comments, I think the title is too general. The authors specifically focus on the baseflow index (BFI) and I believe this should be reflected in the title.

**Reply and revise:** Well taken, we have reflected the baseflow index in the title.

**Comments:** P2, L14: the term "experimental" seems out of place.

**Reply and revise:** We have replaced the term "experimental" with "empirical" (Page 2, Line 14).

**Comments:** P4: what is a "random analysis error" in this context? I suspect the term "analysis" is misused.

**Reply and revise:** Well taken, we have replaced the term "analysis" with "measurement" (Page 4, Line 10).

**Comments:** "conductivity two-component hydrograph separation method" is repeated multiple times: consider introducing an abbreviation.

**Reply and revise**: Well taken, referring to abbreviations in other studies, we have replaced "conductivity two-component hydrograph separation method" with the abbreviation "CMB method" to reduce repetition.

**Comments:** Page 6 and 7: you can simply specify once that the terms W are a standard deviation "multiplied by the t-value ($\alpha=0.05$; two-tail) from the Student's distribution" instead of repeating it each time.

**Reply and revise**: Well taken, we have reduced the repetition in the manuscript as suggested (Page 5, Lines 2--4 & 18--21).

[revised manuscript text omitted]
 $\mathrm{BF_C}$ and $\mathrm{RO_C}$ with that of $\mathrm{BF_C}$ and $\mathrm{RO_C}$, (see Eq. (B1), (B2)):

$$S(\mathrm{BFI}|\mathrm{BF_c}) = \frac{\Delta_{\mathrm{BF_c}} \mathrm{BFI}}{\mathrm{BFI}} \Big/ \frac{\Delta \mathrm{BF_c}}{\mathrm{BF_c}} = \frac{\mathrm{BF_c}(y\mathrm{RO_c} - \sum_{k=1}^{n} y_k SC_k)}{y\mathrm{BFI}(\mathrm{BF_c} - \mathrm{RO_c})^2} \qquad (9)$$

$$S(\mathrm{BFI}|\mathrm{RO_c}) = \frac{\Delta_{\mathrm{RO_c}} \mathrm{BFI}}{\mathrm{BFI}} \Big/ \frac{\Delta \mathrm{RO_c}}{\mathrm{RO_c}} = \frac{\mathrm{RO_c}(\sum_{k=1}^{n} y_k SC_k - y\mathrm{BF_c})}{y\mathrm{BFI}(\mathrm{BF_c} - \mathrm{RO_c})^2} \qquad (10)$$

where $S(\mathrm{BFI}|\mathrm{BF_c})$ represent the dimensionless sensitivity index of BFI (output) with $\mathrm{BF_c}$ (uncertain input), and $S(\mathrm{BFI}|\mathrm{RO_c})$ with $\mathrm{RO_c}$.

The dimensionless sensitivity index is also called the "elasticity index", and it reflects the proportional relationship between the relative error of BFI and the relative error of parameters (e.g. if $S(\mathrm{BFI}|\mathrm{BF_c}) = 1.5$, and the relative error of $\mathrm{BF_c}$ is 5%, then the relative error of BFI is 1.5 times $5\% = 7.5\%$). After determining the values of $\mathrm{BF_C}$, $\mathrm{RO_C}$, BFI, $y$, $y_k$ and $SC_k$, the sensitivity indices $S(\mathrm{BFI}|\mathrm{BF_c})$ and $S(\mathrm{BFI}|\mathrm{RO_c})$ can be calculated and compared.

**2.2 Variables $y_k$ and $SC_k$**

In addition to the two parameters, there are two variables ($SC_k$ and $y_k$) in Eq. (1). This section describes the sensitivity analysis of BFI to these two variables. Similar to Sect. 2.1, the partial derivatives of $b_k$ in Eq. (1) to $SC_k$ and $y_k$ are obtained (see Eq. (A5), (A6)), and the partial derivatives of BFI to $SC_k$ and $y_k$ are further obtained (see Eq. (A7), (A8)),

$$\frac{\partial \mathrm{BFI}}{\partial SC_k} = \frac{1}{\mathrm{BF_c} - \mathrm{RO_c}} \qquad (11)$$

$$\frac{\partial \mathrm{BFI}}{\partial y_k} = \frac{\sum_{k=1}^{n}(SC_k - \mathrm{RO_c}) - n\mathrm{BFI}(\mathrm{BF_c} - \mathrm{RO_c})}{y(\mathrm{BF_c} - \mathrm{RO_c})} \qquad (12)$$

According to previous studies (Munyaneza et al., 2012; Cartwright et al., 2014; Miller et al., 2014; Okello et al., 2018) and this study (Table 1), the difference between $\mathrm{BF_C}$ and $\mathrm{RO_C}$ is often greater than 100 μs/cm. Therefore, $\partial \mathrm{BFI}/\partial SC_k$ is usually less than 0.01 cm/ μs. Appendix C shows that the value of $\partial \mathrm{BFI}/\partial y_k$ is usually far less than 1 d/m$^3$.

Small errors in $SC_k$ and $y_k$ cause errors in BFI

$$\Delta_{SC_k}\text{BFI} = \frac{\partial \text{BFI}}{\partial SC_k}\Delta SC_k = \frac{\Delta SC_k}{\text{BF}_\text{c}-\text{RO}_\text{c}} \tag{13}$$

$$\Delta_{y_k}\text{BFI} = \frac{\partial \text{BFI}}{\partial y_k}\Delta y_k = \frac{\sum_{k=1}^{n}(SC_k-\text{RO}_\text{c})-n\text{BFI}(\text{BF}_\text{c}-\text{RO}_\text{c})}{y(\text{BF}_\text{c}-\text{RO}_\text{c})}\Delta y_k \tag{14}$$

The errors of BFI caused by $SC_k$ and $y_k$ are summed up to obtain the error of BFI caused by $\sum_{k=1}^{n} SC_k$ and $\sum_{k=1}^{n} y_k$ in the whole time series:

$$\Delta_{\sum_{k=1}^{n} SC_k}\text{BFI} = \sum_{k=1}^{n}\Delta_{SC_k}\text{BFI} = \sum_{k=1}^{n}\frac{\Delta SC_k}{\text{BF}_\text{c}-\text{RO}_\text{c}} = \frac{1}{\text{BF}_\text{c}-\text{RO}_\text{c}}\sum_{k=1}^{n}\Delta SC_k \tag{15}$$

$$\Delta_{\sum_{k=1}^{n} y_k}\text{BFI} = \sum_{k=1}^{n}\Delta_{y_k}\text{BFI} = \sum_{k=1}^{n}(\frac{\sum_{k=1}^{n}(SC_k-\text{RO}_\text{c})-n\text{BFI}(\text{BF}_\text{c}-\text{RO}_\text{c})}{y(\text{BF}_\text{c}-\text{RO}_\text{c})}\Delta y_k) = \frac{\sum_{k=1}^{n}(SC_k-\text{RO}_\text{c})-n\text{BFI}(\text{BF}_\text{c}-\text{RO}_\text{c})}{y(\text{BF}_\text{c}-\text{RO}_\text{c})}\sum_{k=1}^{n}\Delta y_k \tag{16}$$

Wagner et al. (2006) reported that the uncertainty of instruments is usually less than 5% for $SC_k$ (<100 μs/cm) and less than 3% for $SC_k$ (>100 μs/cm). According to Hamilton et al. (2012) streamflow data from USGS are often assumed by analysts to be accurate and precise within ±5% at the 95% confidence interval. In this study, the error ranges of $SC_k$ and $y_k$ are all considered to be ±5%. The errors in $SC_k$ and $y_k$ mainly comprise random measurement analysis errors which mostly follow a normal distribution or a uniform distribution (Huang and Chen, 2011). Considering the mutual offset of random errors, when the time series (n) is sufficiently long, $\sum_{k=1}^{n}\Delta SC_{ck}$ in Eq. (15) and $\sum_{k=1}^{n}\Delta y_k$ in Eq. (16) will approach zero.

The analysis of $\sum_{k=1}^{n}\Delta SC_k$ and $\sum_{k=1}^{n}\Delta y_k$ under different time series (n) and different error distributions (normal distribution or uniform distribution) of a surface water station (USGS site number 0297100) showed that the random errors of daily average conductivity and streamflow have a negligible effect on BFI when the time series is greater than 365 days (See Supplement S1 for detail).

**3 Uncertainty estimation**

**3.1 Previous attempts**

According to previous studies, in the case where a variable $g$ is calculated as a function of several factors $x_1, x_2, x_3, ..., x_n$ (e.g. $g = G(x_1, x_2, x_3, ..., x_n)$) and based on the assumptions that the factors are uncorrelated and have a Gaussian distribution, the transfer equation (also known as Gaussian error propagation) between the uncertainty of the independent factors and the uncertainty of $g$ is:

$$W_g = \sqrt{(\frac{\partial g}{\partial x_1}W_{x_1})^2 + (\frac{\partial g}{\partial x_2}W_{x_2})^2 + \cdots + (\frac{\partial g}{\partial x_n}W_{x_n})^2} \tag{17}$$

where $W_g$, $W_{x_1}$, $W_{x_2}$, and $W_{x_n}$ are the same type of uncertainty values (e.g. all average errors or all standard deviations) for $g$, $x_1$, $x_2$, and $x_n$, respectively. A more detailed description of this equation can be found in Taylor (1982), Kline (1985), and Ernest (2005).

According to Genereux (1998), "While any set of consistent uncertainty (W) values may be propagated using Gaussian error propagation, using standard deviations multiplied by $t$ values from the Student's $t$ distribution (each $t$ for the same confidence level, such as 95%) has the advantage of providing a clear meaning (tied to a confidence interval) for the computed uncertainty would correspond to, for example, 95% confidence limits on BFI".

Based on the above principle, Genereux (1998) substituted Eq. (18) into Eq. (17) to derive the uncertainty estimation equation (Eq. (19)) of the two-component mass balance baseflow separation CMB method:

$$f_{bf} = \frac{SC_k - \text{RO}_\text{c}}{\text{BF}_\text{c} - \text{RO}_\text{c}} \tag{18}$$

$$W_{f_{bf}} = \sqrt{(\frac{f_{bf}}{\text{BF}_\text{c}-\text{RO}_\text{c}}W_{\text{BF}_\text{c}})^2 + (\frac{1-f_{bf}}{\text{BF}_\text{c}-\text{RO}_\text{c}}W_{\text{RO}_\text{c}})^2 + (\frac{1}{\text{BF}_\text{c}-\text{RO}_\text{
[revised manuscript text omitted]

---

## Author Response (AR3)

**Response**

**Comments:** Thank you very much for the revised version of your manuscript. I am glad that you addressed most of the reviewer comments and that you also discussed the potential impact of time-variable soil water conductivity. Unfortunately you ignored my request to include a small sensitivity analysis of the effects this could have.

Please include such a sensitivity analysis by assuming a few different, non-constant values of soil water conductivity and demonstrate to the reader with one or two additional figures how these changes propagate through to your sensitivity estimates and eventually the uncertainty.

**Reply:**

Dear editor,

Thank you very much for handling our manuscript. There are obvious temporal and spatial variations in the magnitude and conductivity of soil flow in real watersheds. To analyze the influence of the changes of soil flow conductivity on the results of sensitivity analysis and uncertainty estimation, it is necessary to determine the magnitude of soil flow. Three-component mass balance method (use two tracers) is usually used to determine the magnitude of soil flow (Stewart et al. 2007), but there are few studies on long-time-series. For the convenience of comparative analysis, we assumed that the high-conductivity baseflow is constant, and the ratio of low-conductivity soil flow to the high-conductivity baseflow is between 0 and 1. Then we analyzed the results of BFI, sensitivity and uncertainty under different ratios. The following is a detailed description of the revision of the manuscript.

**Revise:** We have added the following paragraph and Figure to the manuscript (Page 7, Lines 11—24; Page 14).

[revised manuscript text omitted]